# Molecular-scale visualization of sarcomere contraction within native cardiomyocytes

Laura Burbaum[1,4], Jonathan Schneider[2,4], Sarah Scholze[3], Ralph T. Böttcher [3], Wolfgang Baumeister[2], Petra Schwille [1], Jürgen M. Plitzko [2] & Marion Jasnin [2✉]

Sarcomeres, the basic contractile units of striated muscle, produce the forces driving muscular contraction through cross-bridge interactions between actin-containing thin filaments and myosin II-based thick filaments. Until now, direct visualization of the molecular architecture underlying sarcomere contractility has remained elusive. Here, we use in situ cryo-electron tomography to unveil sarcomere contraction in frozen-hydrated neonatal rat cardiomyocytes. We show that the hexagonal lattice of the thick filaments is already established at the neonatal stage, with an excess of thin filaments outside the trigonal positions. Structural assessment of actin polarity by subtomogram averaging reveals that thin filaments in the fully activated state form overlapping arrays of opposite polarity in the center of the sarcomere. Our approach provides direct evidence for thin filament sliding during muscle contraction and may serve as a basis for structural understanding of thin filament activation and actomyosin interactions inside unperturbed cellular environments.

[1] Department of Cellular and Molecular Biophysics, Max Planck Institute of Biochemistry, Martinsried, Germany. [2] Department of Molecular Structural Biology, Max Planck Institute of Biochemistry, Martinsried, Germany. [3] Department of Molecular Medicine, Max Planck Institute of Biochemistry, Martinsried, Germany. [4] These authors contributed equally: Laura Burbaum, Jonathan Schneider. ✉email: jasnin@biochem.mpg.de

Muscle cells contain numerous myofibrils composed of adjoining micrometer-sized contractile units called sarcomeres[1]. These large macromolecular assemblies feature thin filaments, made of polar actin filaments (F-actin) in complex with troponin (Tpn) and tropomyosin (Tpm), and bipolar thick filaments consisting of a myosin tail backbone decorated with myosin-II heads on either side of a head-free region (the so-called bare zone)[1]. The sliding filament theory provided a framework for understanding the mechanism of muscle contraction based on the relative sliding between the myofilaments via cyclic interactions of myosin heads with F-actin[2–5]. Five decades after these pioneering studies much remains to be discovered at the structural level about how the molecular players of the actomyosin machinery cooperate within sarcomeres to produce contractile forces.

Muscle research has benefited greatly from the development of powerful X-ray sources and electron microscopes, allowing detailed views into muscle organization[6], time-resolved X-ray diffraction studies of contracting muscle[7,8], and three-dimensional (3D) reconstruction of actomyosin interactions[9,10]. Thick filaments are known to be crosslinked in the central M-line and occupy the A-band[1,11,12] (Fig. 1a). On either side of the M-line, interdigitating myofilaments organize into hexagonally packed arrays[1,13–15]. Thin filaments extend throughout the I-band toward the Z-disk in which their barbed ends are anchored and crosslinked by α-actinin[1,11,12,16,17] (Fig. 1a). At the molecular level, $Ca^{2+}$-binding to Tpn is thought to trigger a shift in the azimuthal position of Tpm on F-actin to uncover the myosin binding site which allows actomyosin interactions[18,19]. Yet, none of the approaches used so far has permitted molecular-scale imaging of sarcomere organization inside an unperturbed cellular environment.

Recent advances in cryo-electron tomography (cryo-ET) provide access to the 3D molecular architecture of the cellular interior[20,21]. Here, we use state-of-the-art cryo-ET methodologies to unravel sarcomere organization in frozen-hydrated neonatal rat cardiomyocytes across scales, from the myofilament packing to the sliding and functional states of the thin filaments enabling contraction.

## Results and discussion

**Visualizing the myofibrillar interior in situ.** Cardiomyocytes isolated from neonatal rat ventricles provide a valuable experimental system for studying the organization of cardiac sarcomeres during postnatal development. They adapt rapidly to culture conditions[22] and exhibit spontaneous rhythmic contractions after two days (Supplementary Movie 1). This is not the case for cultured adult rat cardiomyocytes, which undergo a process of degeneration and regeneration of their myofibrillar apparatus[23,24]. To explore the microscale organization of myofibrils mediating cell contractility, neonatal rat cardiac myocytes were immunolabelled for Tpn T (one of the three subunits of the Tpn complex), the heavy chain of cardiac myosin and α-actinin, as markers for the thin filament, the thick filament, and the Z-disk, respectively. Myofibrils align into bundles along the principal axes of the star-shaped cells and can split into branches[25] (Fig. 1b–e). Sarcomere branching has been observed in all striated muscle cells, with the frequency of branching regulated by the developmental stage and muscle type[25]. Neonatal myofibrils feature regularly spaced Z-disks and A-bands, which are aligned transversally with those of adjoining myofibrils, with a sarcomere length of $1.8 \pm 0.2$ µm (Fig. 1f, g). In freshly isolated adult mouse cardiomyocytes, myofibrils are more compactly arranged and occupy a tenfold volume resulting from the hypertrophic growth of the adult cells in vivo[26] (Supplementary Fig. 1).

Unlike adult cardiomyocytes, the cell thickness of the neonatal cells is suitable for cryofixation by plunge-freezing (Supplementary Fig. 2a, b). Neonatal rat cardiomyocytes were cultured on electron microscopy (EM) grids and plunge-frozen during spontaneous contraction (Supplementary Fig. 2b). 100- to 200-nm-thick vitrified cellular sections (so-called lamellas) were prepared using cryo-focused ion beam (cryo-FIB) milling[27] (Supplementary Fig. 2c). Transmission electron microscopy (TEM) images revealed cytoplasmic regions filled with long, partially branched myofibrils surrounded by mitochondria and sarcoplasmic reticulum (Fig. 2a and Supplementary Fig. 2d). These myofibrils have a typical diameter of $470 \pm 60$ nm, which corresponds to the smallest diameters observed in cardiac muscle of adult rats (where values can exceed 2 µm)[28] and other mammals[29,30]. They display periodic, electron-dense lines that are aligned with those of neighboring myofibrils. The periodicity is comparable to the sarcomere length, suggesting that these lines are Z-disks. Thus, as observed in the immunofluorescence data (Fig. 1b–g), the sarcomeres of adjacent myofibrils are roughly aligned, even after branching (Fig. 2a). This myofibrillar organization, interspersed with mitochondria and sarcoplasmic reticulum, is reminiscent of that observed in adult rat cardiac muscle using EM[28,31,32].

Myofibrils identified in the lamellas were imaged in 3D by cryo-ET (Supplementary Table 1). The reconstructed volumes permitted visualization of the thin and thick filaments, with diameters of $8.4 \pm 0.9$ and $17.4 \pm 3.0$ nm, respectively (Fig. 2b–d and Supplementary Fig. 3a–c). Densities resembling the head domains of myosin II[10] bridge the space between interdigitating myofilaments (Fig. 2d). Macromolecular complexes of $34.4 \pm 4.3$ nm in diameter, most likely glycogen granules[33], are found outside and within the myofibrillar interior, whereas mitochondria, sarcoplasmic reticulum, and ribosomes are found exclusively in the cytoplasmic space devoid of myofibrils (Fig. 2b, c and Supplementary Fig. 3a–c).

The periodic, electron-dense lines observed in the TEM images are made of irregular, densely packed structures that intersect with the thin filaments in the absence of thick filaments, confirming that they are Z-disks (Fig. 2b, c and Supplementary Fig. 3a–c). Instead of forming a continuous straight line orthogonal to the main axis of the myofibril, they are typically made of two segments of different orientations arranged end-to-end (see the zig–zag appearance, Fig. 2b, c and Supplementary Fig. 3a–c). This raises the possibility that these segments may be associated with different sub-myofibrils that have merged laterally to form a wider, mature myofibril. Similar zig–zag-shaped Z-disks, with more end-to-end segments, have been observed in adult mammalian cardiac muscle[31]. We speculate that the increase in myofibril width during postnatal cardiac development may occur, at least in part, through lateral fusion of sub-myofibrils. This has also been proposed for the transition of nascent myofibrils to mature striated myofibrils in cultured embryonic mouse cardiomyocytes[34,35].

Filament segmentation provided a first glimpse at the native myofilament organization in 3D, revealing regions devoid of thick filaments, corresponding to the I-bands, and a gap between the thin filaments at the location of the Z-disk (Fig. 2e–g, Supplementary Fig. 3d–l and Supplementary Movie 2). This gap results from the high density of the Z-disk structure, which limits the segmentation of the thin filament ends. Neonatal I-bands have widths between 240 and 300 nm, corresponding to 12–19% of the sarcomere length, which is within the range of 10–35% reported in adult mammalian cardiac muscle[31]. Z-disks have widths between 100 and 140 nm, similar to those found in adult vertebrate cardiac muscle[36,37]. The irregularity and high density of the Z-disks in the neonatal cardiac myocytes prevented further structural analysis of this region and the I-bands. Recently, cryo-ET has been used to study the Z-disk and I-band structures in single myofibrils isolated from adult vertebrate cardiac and skeletal muscles[30,38]. Their organization was found to be less ordered than that observed in whole muscles. This is consistent

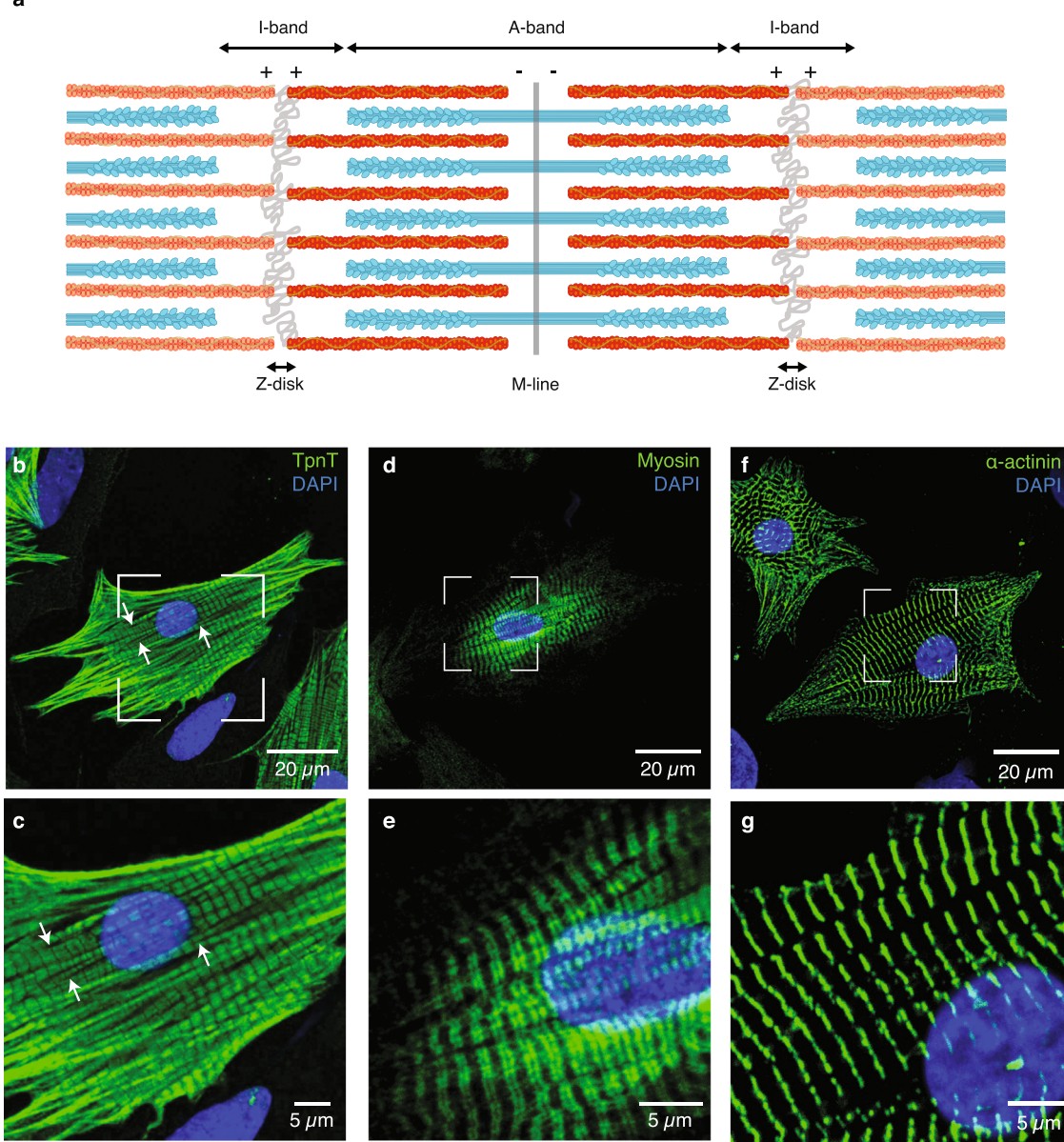

**Fig. 1 Myofibrillar organization in neonatal rat cardiomyocytes at the microscale. a** Schematic representation of adjoining micrometer-sized sarcomeres within a myofibril, showing the organization of the thick (cyan) and thin (red) filaments in this basic contractile unit of striated muscle. Neonatal rat cardiomyocytes were labeled with antibodies specific for troponin T (TpnT; **b**, **c**), the heavy chain of myosin (Myosin; **d**, **e**), and α-actinin (**f**, **g**), respectively. Thin (**b**, **c**) and thick (**d**, **e**) filaments assemble into myofibrils, which align along the main axes of the star-like shaped cells and display regularly spaced Z-disks (**f**, **g**). Myofibril branching is indicated with white arrows. **c**, **e**, **g** Zoomed-in views of the framed regions in **b**, **d**, **f**, respectively. A minimum of three biologically independent experiments were performed in each case.

with the loss of stabilization provided by the myofibrillar matrix, which is connected across both the length and width of the muscle cell during all stages of development[25].

**Excess of thin filaments outside the trigonal positions in the A-bands**. This led us to focus on the A-band organization, where myofilaments were well segmented and are known to be more ordered in adult vertebrate striated muscle[13–15,31]. Interestingly, we did not observe any myosin bare zone depleted of thin filaments, suggesting that these sarcomeres may comprise double-overlap regions (Fig. 2f, g, Supplementary Fig. 3h, i, k, l, and Supplementary Movie 2). We quantitatively analyzed the 3D organization of the neonatal cardiac myofilaments in the single-overlap regions using an approach described previously[39] ("Methods"). As shown by the six-fold symmetry in Fig. 3a, thick

filaments organize in a hexagonal lattice with an interfilament distance of 45.1 ± 3.8 nm (Fig. 3a–c and Supplementary Fig. 4a, b). Given the sarcomere lengths measured in our cells, these values are in good agreement with the spacings found in adult mammalian heart muscle using X-ray diffraction[40,41]. The roughly three-fold symmetry in Fig. 3d indicates that the thin filaments are found at the trigonal positions of the lattice, with a distance of 26.0 ± 2.4 nm from the thick filaments (Fig. 3d–f and Supplementary Fig. 4c–f). Combined together, these spacings agree precisely with the geometrical constraints of a double hexagonal lattice. However, the mean distance of 15.5 ± 1.4 nm between nearest thin filaments indicates that thin filaments are also present outside the trigonal positions of the array (Supplementary Fig. 4g, h). Therefore, the packing of the thin filaments around the thick filaments we observed, both in real cross-sections and

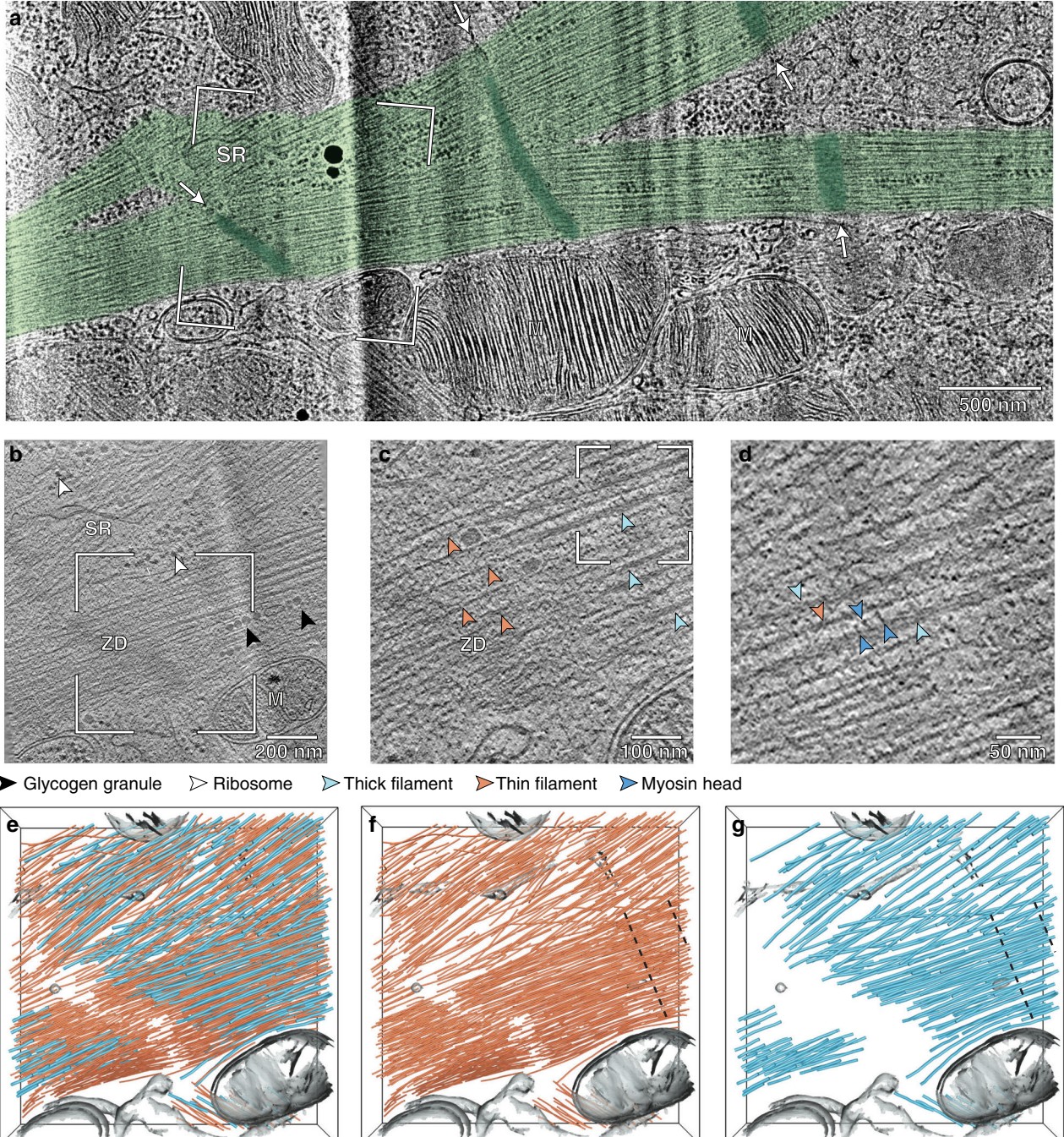

**Fig. 2 Visualizing the myofibrillar interior in situ. a** TEM image of the enlarged framed area of a neonatal rat cardiomyocyte lamella shown in Supplementary Fig. 2d. Myofibrils (green) display periodic, electron-dense lines (dark green, white arrows) aligned transversally with those of their neighbors, even after branching. M mitochondrion, SR sarcoplasmic reticulum. **b** 6.84 nm-thick slice from a defocused tomographic volume acquired at the framed area in **a** showing the myofibrillar architecture. ZD Z-disk. Ribosomes and glycogen granules are indicated by white and black arrowheads, respectively. The tomogram has been rotated by 90° so that the TEM and tomographic images have the same orientation. The original tomogram has been deposited in the EMDB under accession code EMD-12572. **c** Zoomed-in view of the framed area in **b** showing a zig-zag-shaped Z-disk intersecting with thin filaments (orange arrowheads) in the absence of thick filaments (cyan arrowheads). **d** Zoomed-in view of the framed area in **c** showing myosin heads (dark blue arrowheads) bridging the myofilaments. **e–g** 3D rendering of the cellular volume shown in **b** revealing the nanoscale organization of thin (orange; **e**, **f**) and thick (cyan; **e**, **g**) filaments within unperturbed sarcomeres. The putative myosin bare zone estimated for an average sarcomere length of 1.8 μm is delineated by dashed black lines in **f**, **g**. See also Supplementary Movie 2. Additional examples are provided in Supplementary Fig. 3. A total of 13 tomograms acquired from eight cells were used in this study (Supplementary Table 1). Each cell is from two separate experiments in which four organisms were mixed and can therefore be considered as a biological replicate. Representative images are shown.

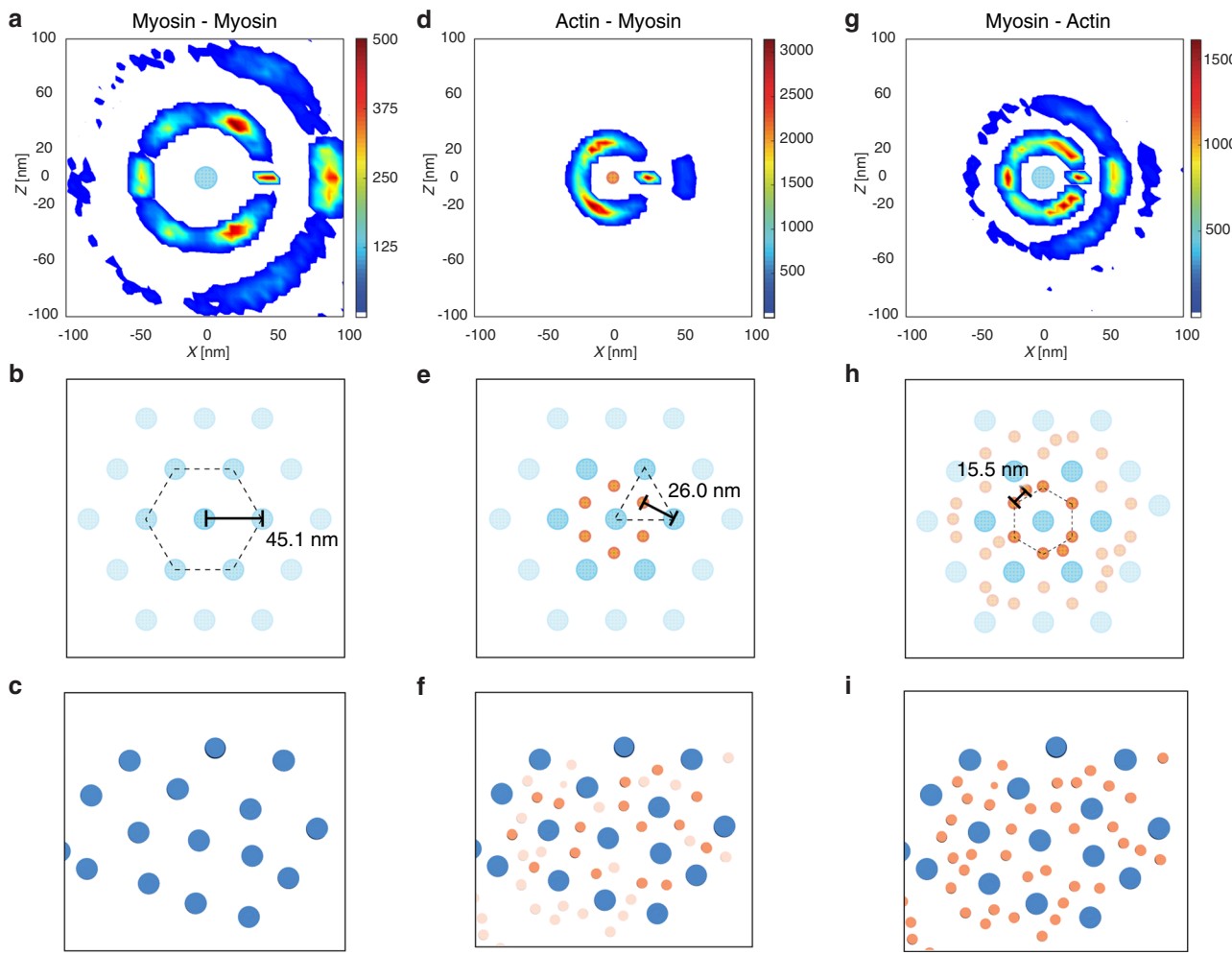

**Fig. 3 Neonatal cardiac thick filaments are hexagonally packed with an excess of thin filaments outside the trigonal positions. a, d, g** Quantitative analysis of myofilament packing in neonatal rat cardiomyocytes, **b, e, h** the schematic representation of the idealized arrays, and **c, f, i** the corresponding cross-sections through the single-overlap region of the myofibril shown in Supplementary Fig. 3e, k. See Supplementary Fig. 5 for additional cross-section examples. In short, the analysis consists of searching for parallel neighboring filaments within the first and second shells around the filaments, using the distance ranges determined for the nearest neighbors (Supplementary Fig. 4). The plots correspond to the heat maps of neighbor positions around thick filaments (myosin, cyan-filled circle; **a**, **g**) or thin filaments (actin, orange-filled circle; **d**) in a plane perpendicular to the filament cross-sections. Since neighbors are mainly found within the first shell, the second shell was blurred in **b**, **e**, **h**. In addition, thin filaments outside the trigonal positions in **f** were also blurred to facilitate comparison with the idealized array in **e**. Thick filaments assemble into a hexagonal lattice with an interfilament distance of about 45.1 nm (**a–c**, Supplementary Fig. 4a, b). Thin filaments are found at about 26.0 nm from three neighboring thick filaments (**d–f**, Supplementary Fig. 4c–f). They are also located outside the trigonal positions of the thick filament lattice (**g–i**), as indicated by the more even distribution of density in the first shell in **g** and the distance of about 15.5 nm between nearest parallel thin filaments (Supplementary Fig. 4g, h).

quantitatively, is less ordered than that found in adult vertebrate striated muscles[15,31,40,42–44] (Fig. 3g–i and Supplementary Fig. 5). This is consistent with the thin-to-thick filament ratio of 3:1 found in our neonatal rat cardiomyocytes, as compared to the ratio of 2:1 reported in adult vertebrate skeletal and cardiac muscles[31,45], which share the same myofibril ultrastructure[31]. Given that these ratios are greater than 7:1 in embryonic mammalian skeletal muscle[46], our data indicate that upon vertebrate muscle development, the number of thin filaments and their packing around thick filaments are progressively optimized to provide the double hexagonal array observed at the adult stage. In addition, our data show that sarcomere contraction can occur despite the higher density and reduced order of the thin filaments around the hexagonally packed thick filaments.

**Structural signature of sarcomere contraction.** Next, we exploited subtomogram averaging[47] to unveil the molecular

organization of the thin filaments enabling sarcomere contraction. To this end, we first developed an approach to obtain the structure of the thin filaments by aligning and averaging multiple consecutive actin subunits along the filaments without imposing any helical symmetry ("Methods"). Thin filaments located on one side of a Z-disk were used to generate a de novo structure of F-actin in a complex with Tpm resolved at 20.7 Å (Supplementary Fig. 6a–c). At this resolution, the positions of the actin subunits with respect to the two Tpm strands allow to discriminate between F-actin structures of opposite polarity (Supplementary Fig. 6d, e). Comparison with a pseudoatomic model of the F-actin-Tpm complex[10] permitted to allocate the barbed (+) and pointed (–) ends of the actin filament (Fig. 4a and Supplementary Fig. 6f, g).

We then sought to determine the polarity of the thin filaments within the neonatal cardiac sarcomeres. Using a multireference alignment procedure based on the obtained de novo structure, we

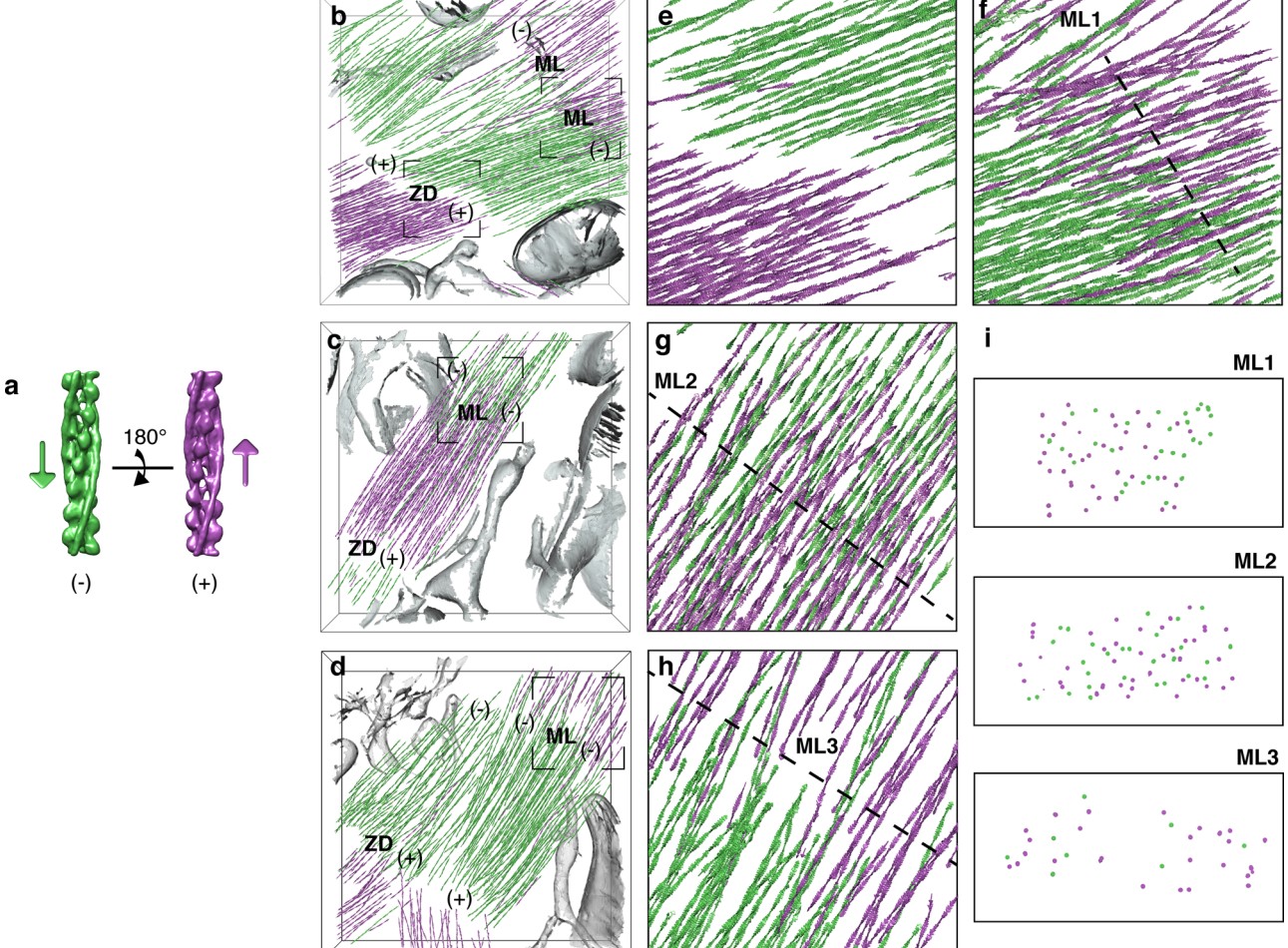

**Fig. 4 Structural signature of sarcomere contraction. a** De novo structures of the actin-Tpm filament in opposite orientations, represented by arrows pointing toward the pointed (−) end. **b**−**d** Polarity assignment for the neonatal cardiac thin filaments shown in Fig. 2e, f and Supplementary Movie 2 (**b**; Supplementary Movie 3), Supplementary Fig. 3e, h (**c**; first half of Supplementary Movie 4) and Supplementary Fig. 3f, i (**d**; second half of Supplementary Movie 4). Filaments are represented by arrows colored according to the assigned polarity. The gaps between thin filaments facing each other with their barbed ends indicate the location of Z-disks (ZD). The overlap regions between thin filaments of opposite polarity at the pointed (−) ends correspond to M-lines (ML). **e**, **f** Zoomed-in views of the framed regions in **b** showing the thin filament orientations at the Z-disk (**e**) and M-line (**f**, ML1). **g**, **h** Zoomed-in views of the framed area in **c**, **d**, respectively, showing the thin filament orientations at the M-lines (ML2−3). **i** Cross-sections through ML1−3 at the locations indicated by the dotted lines in **f**−**h**. See Supplementary Fig. 8 for visualization of the unassigned filaments.

evaluated the polarity of each individual filament from a series of statistically independent estimates ("Methods" and Supplementary Fig. 7a, b). 69% of the data were assigned a polarity with confidence (Supplementary Fig. 7c) and visualized in 3D (Fig. 4 and Supplementary Fig. 8). Filaments belonging to the same bundle shared the same polarity, while changes of polarity were observed between adjoining bundles (Fig. 4b–d and Supplementary Fig. 8a–c). In the vicinity of the Z-disks, thin filaments face each other with their barbed ends (Fig. 4e and Supplementary Movie 3), as has been reported for embryonic and adult mammalian skeletal muscles using heavy meromyosin decoration of thin filaments[12,16]. In addition, the analysis showed regions in which thin filaments of opposite polarity face each other with their pointed ends, revealing the location of M-lines (Fig. 4f–h, ML1–3, and Supplementary Fig. 8d–f).

This allowed us to uncover the arrangement of the thin filaments in the center of the sarcomere, where arrays of opposite polarity are thought to be pulled toward each other during contraction. In two out of three M-lines, thin filaments of opposite polarity substantially overlap with each other, revealing

their relative sliding (Fig. 4f–g, ML1–2, Supplementary Fig. 8d, e and first half of Supplementary Movie 4). Based on length–tension measurements in adult frog skeletal muscle fibers, Gordon et al.[48] have proposed that as the sarcomere length decreases below 2 μm, the thin filaments on either side of the M-line begin to overlap. This was later inferred in adult frog and rat heart muscles[49]. In the sarcomere shown in Fig. 4c, the overlap length is 281 nm, corresponding to 17% of the sarcomere length. This is twice the length of 140 nm reported for the bare zone (so-called M-region) of mammalian cardiac muscles[50], indicating that these thin filaments slid past the M-region. The sarcomere length of 1.65 μm is smaller than the average value of 1.8 ± 0.2 μm we measured in these cells, in agreement with sarcomere shortening. Furthermore, this length corresponds to that estimated for an overlap that extends beyond the M-region in adult frog skeletal muscle[48]. Cross-sections through the M-lines show that thin filaments of opposite polarities can contribute to the packing around the same thick filaments during their sliding (Fig. 4i, ML1–2, and Supplementary Fig. 8g, h). We did not observe a significant increase in the number of thin filaments in these

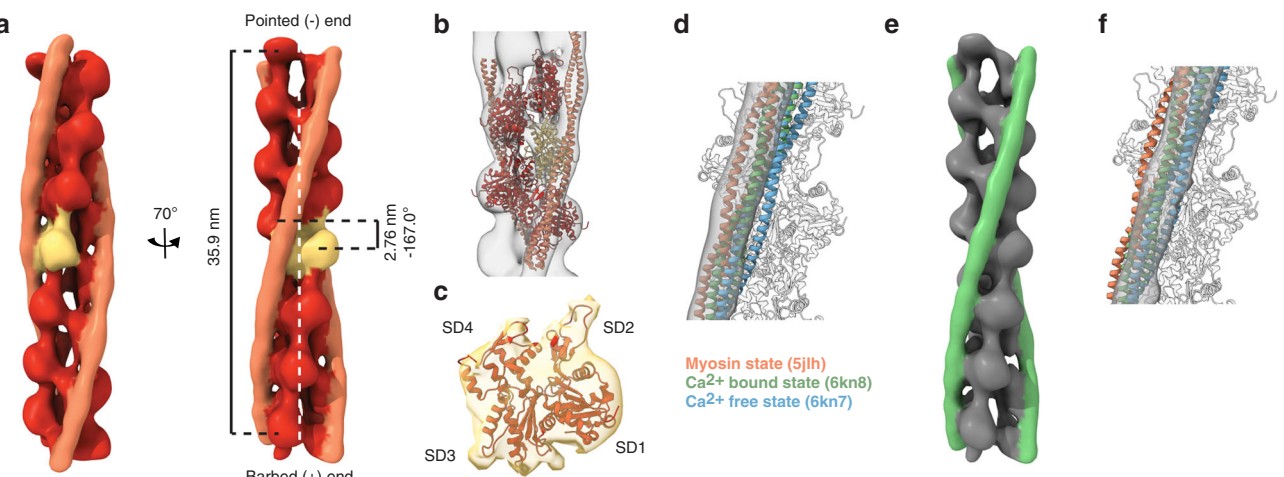

**Fig. 5 In situ structure of the cardiac thin filament in two distinct functional states. a** In situ subtomogram average of the cardiac actin filament (red and yellow) in complex with Tpm (orange) resolved at 15.7 Å (Supplementary Fig. 9c, orange curve, and Supplementary Movie 5). This structure was obtained from the wide-overlap data (Fig. 4b, c). The fit of a pseudoatomic model of the F-actin-Tpm complex[10] into the subtomogram average (**b**) reveals F-actin polarity at the level of the actin monomers (**c**). **d** Docking of pseudoatomic models in different states[10,19] into the map reveals that the thin filament is in the myosin state (see also Supplementary Fig. 9d). **e** In situ subtomogram average of the cardiac actin filament (gray) in complex with Tpm (green) resolved at 17.7 Å (Supplementary Fig. 9c, green curve). This structure was obtained from the narrow-overlap data (Fig. 4d). **f** The Ca$^{2+}$ bound state fitted best to the subtomogram average in **e** (see also supplementary Fig. 9e), indicating that this sarcomere is in a different state of the contraction cycle.

double-overlap regions. This could be explained by the variation in thin filament length within a single sarcomere in cardiac muscle, which provides incomplete shells of thin filaments around thick filaments near the M-region[49]. In the sarcomere shown in Fig. 4d, the overlap between thin filaments of opposite polarity is barely visible, indicating that this sarcomere may be in a different contraction state (Fig. 4h, i, ML3, Supplementary Fig. 8f, i and second half of Supplementary Movie 4). We measured an overlap length of 60 nm for a sarcomere length of 1.96 µm, i.e., an overlap region of only 3% of the sarcomere length. This agrees with the onset of thin filament overlap estimated for a sarcomere length of about 2 µm in adult frog skeletal muscle fibers[48].

**In situ structure of the cardiac thin filament in two distinct functional states.** We next used subtomogram averaging to determine the structures of the thin filaments associated with the different observed M-line organizations. We first analyzed each tomogram individually in order to uncover potential differences in the Tpm position between sarcomeres ("Methods"). The two tomograms, where thin filaments of opposite polarity substantially overlap in the M-line region, provided a similar F-actin-Tpm structure (Supplementary Fig. 9a, blue (ML1) and dark blue (ML2) Tpm densities, ML1/2). In contrast, the data shown in Fig. 4d yielded a subtomogram average in which the Tpm density is shifted azimuthally on the surface of the actin filament (Supplementary Fig. 9a, cyan Tpm (ML3), ML1/3–2/3), indicating that these filaments are in a different functional state.

To correlate the wide overlap observed at the M-lines in Fig. 4b, c (ML1–2) with the structure of the thin filament, the particles from these two tomograms were combined. A refined subtomogram average of the cardiac thin filament was obtained at a resolution of 15.7 Å (Fig. 5a, Supplementary Fig. 9b, c (orange Tpm/curve), Supplementary Movie 5, and Supplementary Table 2). The structure features 13 fully resolved F-actin subunits surrounded by two Tpm strands which approximately follow the

same helical twist (Fig. 5a and Supplementary Movie 5). The actin filament consists of a left-handed single helix with a full repeat distance of 35.9 nm, a rise per actin subunit of 2.76 nm, and an angular twist per molecule of –167.0°, in agreement with the literature[51]. The resolution allowed the docking of a pseudoatomic model of the F-actin-Tpm complex into the map (Fig. 5b and Supplementary Movie 5). F-actin polarity is evident at the level of the actin monomers (Fig. 5c and Supplementary Movie 5). Comparison with pseudoatomic models in different functional states[10,19] showed that the thin filament is in the fully activated ("myosin") state that allows strong binding of the myosin heads (Fig. 5d and Supplementary Fig. 9d). This provides evidence at the molecular level that a majority of thin filaments in these sarcomeres are strongly bound to thick filaments, allowing their relative sliding and, thus, sarcomere contraction. Moreover, the Ca$^{2+}$ bound state of the cardiac thin filament, which allows myosin head access but no strong binding[18], fitted best to the separate structure obtained from the data shown in Fig. 4d (Fig. 5e, f, Supplementary Fig. 9b, c (green Tpm/curve), e and Supplementary Table 2). This supports that this sarcomere is in a different state of the contraction cycle, in agreement with the distinct thin filament organization observed at the M-line, where arrays of opposite polarity barely overlap (Fig. 4d, h).

In summary, our integrative approach permitted us to directly visualize sarcomere organization in pristinely preserved neonatal cardiomyocytes with molecular precision. Our work provided structural insights into a specific stage of cardiac development, without any disturbance caused by myofibril isolation, decoration, or staining, while supporting past EM and X-ray diffraction studies of adult vertebrate striated muscles. At the neonatal stage, contraction occurs with an excess of thin filaments outside the trigonal positions of the thick filament hexagonal lattice. We showed that, despite this reduced order of the thin filaments, neonatal cardiomyocytes are a valuable system for structural analysis of sarcomere contraction. Subtomogram averaging revealed that the polarity of the cardiac thin filaments is

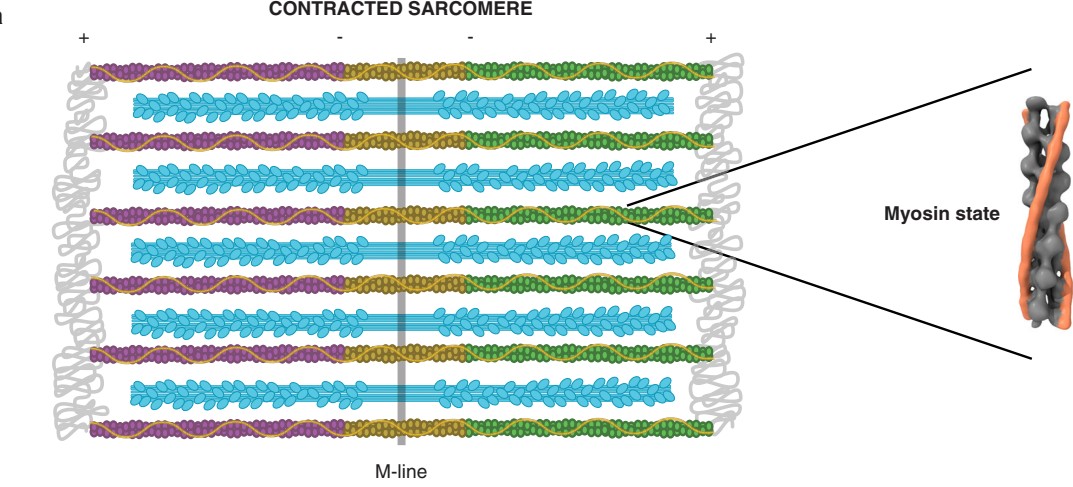

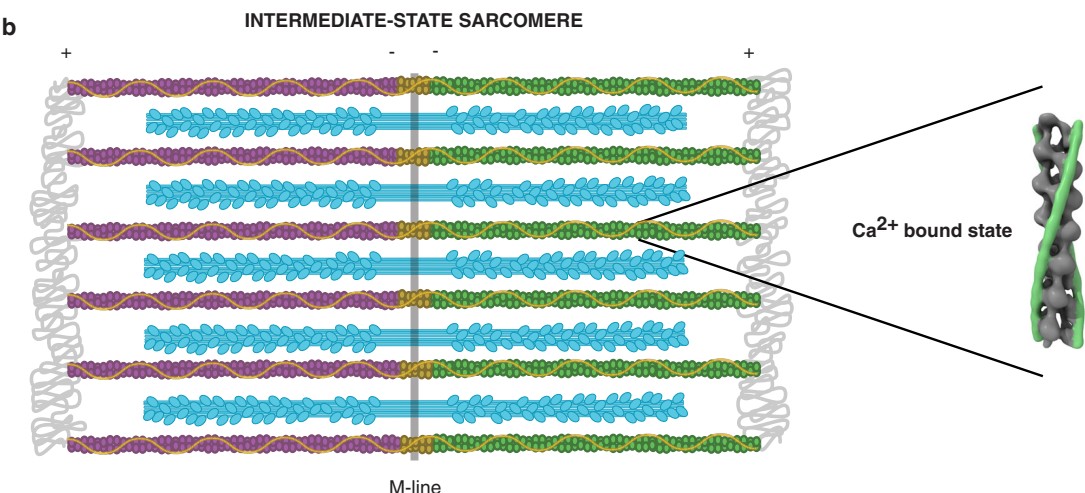

**Fig. 6 Thin filament organization during sarcomere contraction in neonatal cardiomyocytes. a** Contracted sarcomeres in which thin filaments are strongly bound to thick filaments, exhibit an overlap between thin filaments of opposite polarity beyond the bare zone region. **b** When the actomyosin interaction is weaker, sarcomeres are found in an intermediate state in which thin filaments of opposite polarity barely overlap.

already established at the neonatal stage. In addition, contracted sarcomeres in which thin filaments are strongly bound to thick filaments, exhibit an overlap between thin filaments of opposite polarity in the M-region, confirming that they slide past each other during contraction (Fig. 6a). When the actomyosin binding is weaker, sarcomeres are found in an intermediate state in which thin filaments of opposite polarity barely overlap (Fig. 6b). Further in situ studies are required to explore the extent to which this overlap occurs in other muscle types and development stages. Our work demonstrates that cryo-ET is able to reveal the polarity and functional states of the thin filaments in situ, and paves the way for structure determination of actomyosin interactions within cells.

## Methods

**Isolation of ventricular cardiomyocytes from neonatal rats and adult mice.** The rats and mice used in this study were housed under specific pathogen-free conditions at room temperature (RT; 20–24 °C) and 45–65% humidity. They had continuous access to drinking water and standard rodent food. The light regime was set at 12 h of light per day. The housing and use of laboratory animals at the Max Planck Institute of Biochemistry are fully compliant with all applicable German (e.g., German Animal Welfare Act) and EU (e.g., Annex III of Directive 2010/63/EU on the protection of animals used for scientific purposes) laws and regulations regarding the care and use of laboratory animals. All animals were handled in accordance with the approved license (No.5.1-568—rural districts office).

Neonatal rat ventricular cardiomyocytes were isolated from 3-day-old (P3) wild-type Wistar rats of both sexes. The rats were killed by decapitation and the ribcage was opened. The heart was removed by pulling with forceps and transferred into a 35-mm Petri dish containing phosphate-buffered saline (PBS). The remaining blood was pumped out of the heart and the great vessels. Both atria and the connective tissue were removed. For each preparation, the ventricles from four hearts were mixed, cut into small pieces, and digested using the Neonatal Heart Dissociation kit (Miltenyi Biotec, Bergisch Gladbach, Germany). Briefly, the enzyme mix 1 was pre-heated for 5 min at 37 °C and added to the enzyme mix 2. The harvested and chopped ventricular tissue was transferred to the tube containing the enzyme mix. The suspension was gently pipetted every 10 min for 1 h. Following enzymatic digestion, the cells were passed through a 70-μm cell strainer and centrifuged at 300 g for 15 min. The pellet was resuspended in 10 mL of cardiomyocyte medium. This medium consists of a 1:1 mixture of Dulbecco's modified Eagle medium and Ham's F12 medium (Life technologies, #31330-038) containing 5% horse serum, 5% fetal calf serum, 20 μM Cytarabine (Ara-C) (Sigma Aldrich, #C1768), 3 mM sodium pyruvate (Sigma #S8636), 2 mM L-glutamine (Thermo Fisher Scientific, #25030-081), 0.1 mM ascorbic acid (Sigma #A4034), 1:200 insulin–transferrin–selenium–sodium pyruvate (Invitrogen, #51300044), 0.2% bovine serum albumin (BSA) (Sigma Aldrich, # A7409) and 100 U/mL penicillin-streptomycin (Thermo Fisher Scientific, #15140122). The resuspended pellet was pre-plated onto a non-coated cell culture dish for 90 min to reduce the number of cardiac fibroblasts. The supernatant was transferred to a 15 mL tube and centrifuged at 300 g for 15 min. The pellet containing the neonatal rat ventricular

cardiomyocytes from four hearts was resuspended in 4 mL of cardiomyocyte medium (that is, 1 heart eq./mL).

Primary adult ventricular cardiomyocytes were isolated from 12- to 15-week-old wild-type C57BL/6 male mice following the protocol from Ackers-Johnson et al.[52]. Briefly, the mice were killed by $CO_2$. Immediately after the absence of toe-pinch reflex, the mice were transferred to the surgical area and fixed. The thorax was wiped with 70% ethanol and opened. 7 mL ethylenediaminetetraacetic acid (EDTA) buffer was injected into the base of the right ventricle, while the descending aorta was cut. The heart was then removed and transferred to a dish containing EDTA buffer. A series of injections (EDTA, perfusion, and collagenase) was applied to the left ventricle and the heart was placed in dishes containing the same buffers. Once digestion was complete, the atria were removed and the ventricles were placed in a separate dish. Trituration with a 1 mL pipette allowed further digestion of the tissue. Digestion was stopped by adding stop buffer and the cell suspension was transferred through a 100-μm pore-size cell strainer (Corning®, #431752). To remove cell debris and non-myocytes from the cell suspension, the cells were allowed to settle by gravity in four rounds by reintroducing a calcium-containing medium. The final myocyte pellet was resuspended in a plating medium.

**Culture and vitrification of neonatal cardiomyocytes on EM grids.** Gold EM grids with Quantifoil R 2/1 holey carbon film (Quantifoil Micro Tools GmbH, Grossloebichau, Germany) were glow-discharged, sterilized by UV irradiation for 30 min and coated with fibronectin (10 μg/mL, Merck, #341631). 6–8 EM grids were placed in 35-mm Petri dishes containing 1.5 mL of cardiomyocyte medium and 300 μL of cell suspension was added (that is, ~ 1/3 heart eq. per dish). The cells were cultured at 37 °C in 5% $CO_2$ for 2 days. To verify that the cells exhibit spontaneous rhythmic contractions prior to cryo-fixation, the grids were imaged at RT by bright field using a 20× (air, NA 0.8) Plan Achromat objective (Carl Zeiss, Jena, Germany) of a CorrSight microscope (Thermo Fisher Scientific). The grids were plunge-frozen using a Vitrobot Mark IV (Thermo Fisher Scientific) with 80% humidity and 10 s blotting time from the back side in a 2:1 ethane/propane mixture cooled by liquid nitrogen.

**Immunofluorescence microscopy.** Neonatal cells used for immunofluorescence microscopy were cultured on a chambered coverslip with eight wells (IBIDI #80826, Gräfelfing, Germany) coated with fibronectin (or laminin for the adult cells; 10 μg/mL (resp. 5 μg/mL) in PBS at 37 °C for 1 h). After 2 days (or 1.5 h for the adult cells), the cells were rinsed in PBS, placed on ice, and fixed in 4% paraformaldehyde (PFA) for 15 min. After PBS wash, the cells were permeabilized with 0.1% Triton X-100/PBS for 10 min and blocked with 3% BSA/PBS at 4 °C for at least 1 h. Subsequently, the cells were incubated at 4 °C with one of the following mouse monoclonal primary antibodies: anti-heavy chain cardiac myosin (Abcam, #ab50967, 1:100), anti-α-actinin (Sigma-Aldrich, #A7811, 1:200) or anti-cTnT (Thermo Fisher Scientific, #MA5-12960, 1:200). The next day, the cells were washed with PBS (3 × 5 min) and incubated with an anti-mouse secondary antibody (Alexa Fluor™ 488 goat IgG (H + L), Thermo Fisher Scientific, #A11029, 1:600) at RT for 1 h in the dark. The cells were washed with PBS (2 × 5 min), stained with DAPI (1:6,000 in PBS) at RT for 15 min in the dark, washed again in PBS (2 × 5 min), and stored at 4 °C. Confocal imaging was performed with a Zeiss LSM 780 confocal laser scanning microscope (Carl Zeiss, Oberkochen, Germany) equipped with a 40× (oil, NA 1.4) Plan-Apo objective (Carl Zeiss, Jena, Germany). The resulting images were analyzed with ImageJ software version 1.52p (https://imagej.net/Fiji).

**Cryo-FIB milling.** Plunge-frozen grids were clipped into Autogrids modified for cryo-FIB milling[53,54]. The Autogrids were mounted into a custom-built FIB shuttle cooled by liquid nitrogen and transferred using a cryo-transfer system (PP3000T, Quorum Technologies) to the cryo-stage of a dual-beam Quanta 3D FIB/SEM (Thermo Fisher Scientific) operated at liquid nitrogen temperature. During the loading step, the grids were sputter-coated with platinum in the Quorum prep-chamber (10 mA, 30 s) to improve sample conductivity. To reduce curtaining artifacts, they were subsequently sputter-coated with organometallic platinum using the gas injection system (GIS, Thermo Fisher Scientific) in the microscope chamber operated for 8 s at 26 °C. Lamellas were prepared using a Gallium ion beam at 30 kV and stage tilts of 18–20°. 8- to 12-μm wide lamellas were milled in a step-wise manner using high currents of 0.5 nA for rough milling that were gradually reduced to 30 pA for fine milling and final cleaning steps. The milling process was monitored using the electron beam at 5 kV and 11.8 pA or 3 kV and 8.87 pA. For Volta phase plate (VPP) imaging, the lamellas were additionally sputter-coated with a platinum layer in the Quorum prep-chamber (10 mA, 3 s). A total of eight lamellas from randomly chosen cells was used to produce the data presented in the paper.

**Cryo-ET.** The lamellas were loaded in a Titan Krios TEM (Thermo Fisher Scientific) equipped with a 300-kV field-emission gun, VPPs (Thermo Fisher Scientific), a post-column energy filter (Gatan), and a 4k × 4k K2 Summit direct electron detector (Gatan) operated using SerialEM[55]. The VPPs were aligned and used as described previously[56]. Low-magnification images were captured at 6,500×.

High-magnification tilt series were recorded in counting mode at 42,000x (calibrated pixel size of 0.342 nm), typically from –50 to 70° with 2° steps, starting from a pre-tilt of 10° to correct for the lamella geometry, and a total dose of ~ 100–120 e⁻/Å². A total of 13 tomograms was used in this study (Supplementary Table 1). Three of the tilt series, collected from one cell, were recorded uni-directionally with the VPPs at a target defocus of 0.5 μm. Ten of the tilt series, collected from seven different cells, were recorded without the VPP using a dose-symmetric tilt scheme[57] and a target defocus range of –3.25 to –5 μm.

**Tomogram reconstruction.** Preprocessing was performed using the TOMOMAN package (https://github.com/williamnwan/TOMOMAN) as follows: frames were aligned using MotionCor2[58] and dose-filtered by cumulative dose using the exposure-dependent attenuation function and critical exposure constants described in[59] and adapted for tilt series in[60,61]. Tilt series were aligned using patch tracking in IMOD[62] and tomograms were reconstructed with a binning factor of 4 (13.68 Å per pixel) using weighted back-projection. For visualization, tomograms were filtered using a deconvolution filter (https://github.com/dtegunov/tom_deconv). For subtomogram averaging, contrast transfer function (CTF) parameters for each tilt were estimated by CTFFIND4[63]. A 3D-CTF correction was performed using NovaCTF[64] with phase flip correction and a defocus step of 15 nm, and the reconstructed tomograms were binned by a factor of 2 (6.84 Å per pixel).

**Membrane and filament segmentation.** The 4× binned tomograms (pixel size of 13.68 Å) were used for segmentation. Membranes were generated automatically using tomosegmemtv[65] and refined manually in Amira (Thermo Fisher Scientific). Thin and thick filaments were traced automatically in Amira using an automated segmentation algorithm based on a generic cylinder as a template[66] and implemented in the X-Tracing extension. The cylindrical templates were generated with a length of 42 nm and diameters of 8 and 18 nm for the thin and thick filaments, respectively.

**Near-neighbor analysis and local packing analysis.** The analysis developed in[39] was adapted to describe quantitatively the 3D organization of the cardiac myofilaments in the A-bands. We note that the single-overlap regions were not specifically selected. However, given that the polarity analysis revealed only a few double-overlap regions over less than 20% of the A-band length, their contribution to the packing can be considered to be small. A total of 4,770 thin filaments and 1,648 thick filaments extracted from nine tomograms were used (Supplementary Table 1). Data analysis was performed in MATLAB (The MathWorks) using the coordinates of the filament centerlines exported from Amira as input. The coordinates of the filaments were resampled every 3 nm to give the same weight to every point along a filament.

Near-neighbor analysis: the local direction at each point was evaluated as the local tangent of the filament at that point. For every point within a filament, the closest point of each neighboring filament was characterized by its distance, $d$, and relative orientation, $\theta$, with respect to the reference point. The analysis was performed for filaments of the same type (that is, only thick filaments or only thin filaments) as well as for filaments of one type relative to the other (that is, neighboring thin filaments for the thick filaments, and reciprocally). The occurrences of $(d, \theta)$ were represented by a 2D histogram. The peak occurring at small $\theta$ (below 10°) indicates the presence of equidistant and nearly parallel filaments. The mean interfilament spacing was obtained from the distances with a number of occurrences higher than two-third of the peak maximum (Supplementary Fig. 4).

Local packing analysis: a local reference frame (**e1**, **e2**, **e3**) was defined at every point along a filament as follows: **e2** points in the local direction of the filament, **e1** points in the direction of the nearest neighbor, and **e3** is the cross-product of **e1** and **e2** (see[39]). $(X, Y, Z)$ are the coordinates of a neighbor position in (**e1**, **e2**, **e3**). The local reference frames (**e1**, **e2**, **e3**) of all points along the filaments were aligned, and the occurrences of $(X, Z)$ in the **e1e3** plane within the distance range for parallel filaments were represented by a 2D histogram (Fig. 3a, d, g).

**Subtomogram averaging and polarity assessment.** Subtomogram sampling: ten tomograms, acquired with a dose-symmetric tilt scheme and a target defocus range of –3.25 to –5 μm, were used for subtomogram averaging and polarity assessment (Supplementary Table 1). The coordinates of the thin filaments extracted from Amira were resampled every 1.38 nm (corresponding to half of the axial rise per actin subunit in the actin filament) along the filament centerline using scripts described in[39]. This was instrumental to correctly allocate every actin subunit during subtomogram alignment[61]. Principal component analysis was used to generate a common direction for all the filaments in the network upon resampling (that is, filament coordinates were reordered so that the dot product between the direction of each filament and the first principal component was positive). The local tangent at each point along a filament served to calculate initial Euler angles ($\varphi$, $\theta$, $\psi$) in the zxz-convention using scripts from[39] and TOM Toolbox[67]. As a result, the filament centerline was aligned with the z-axis, allowing the $\varphi$ angle to describe the in-plane rotation. Since this rotation cannot be determined directly from the tomogram, $\varphi$ was randomized every 30°. Furthermore, subvolumes

belonging to the same filament were assigned a unique identifier, which was used during the assessment of filament polarity exclusively.

*De novo reference:* subtomogram averaging was performed using STOPGAP[68]. An initial reference was generated de novo from a single tomogram as follows. 84 thin filaments located on one side of a Z-disk and belonging to the same bundle (shown in orange in Supplementary Fig. 6a) were selected, based on the assumption that they had the same polarity. 20,259 subtomograms were extracted from the resampled positions using a binning factor of 2 (6.84 Å per pixel) and a box size of $128^3$ pixels$^3$ and a starting reference was generated by averaging all subvolumes. Due to the in-plane randomization, the initial structure resembles a featureless cylinder (Supplementary Fig. 6b). Several rounds of global alignment of the in-plane angle followed by local refinement of all Euler angles were conducted. Shift refinements were limited to 1.38 nm in each direction along the filament centerline. This averaging approach resulted in the emergence of the helical pattern of the thin filament (Supplementary Fig. 6b). Unbinned subvolumes were extracted using a box size of $128^3$ pixels$^3$ and aligned iteratively, resulting in a refined structure of F-actin in complex with Tpm at a resolution of 20.7 Å (Supplementary Fig. 6b, c). At this resolution, the positions of the actin subunits with respect to the two Tpm strands allowed to distinguish between F-actin structures of opposite polarities (Supplementary Fig. 6d, e). Comparison with a pseudoatomic model of the F-actin-Tpm complex[10] permitted to allocate the barbed (+) and pointed (−) end of the de novo F-actin structure (Supplementary Fig. 6f, g).

*Polarity assessment:* the 20.7 Å structure of F-actin in complex with Tpm was rotated by 180° around the *x*-axis to generate a second structure with opposite polarity (Fig. 4a and Supplementary Fig. 6d). Both structures were used to assess the polarity of the thin filaments in each tomogram using multireference alignment (Supplementary Fig. 7a). Each subvolume within a tomogram was extracted with a binning factor of 2 (6.84 Å per pixel) using a box size of $64^3$ pixels$^3$ and aligned against both references independently. The scores of all the subtomograms belonging to the same filament were compared using unpaired Student's *t*-test (Supplementary Fig. 7b). On a total of 4,438 filaments (corresponding to 594,317 subvolumes) from ten tomograms, 69% of the data showed significant differences between the two directions ($p < 0.05$) and were assigned the respective polarity with confidence (Supplementary Fig. 7c). The allocated filaments were kept for further processing. Polarity assignment was visualized with arrows pointing in the direction of the pointed ends of F-actin using the Place Object tool in UCSF Chimera[69,70] (Fig. 4b–h and Supplementary Fig. 8a–f).

*Subtomogram averaging of the thin filament:* to generate a dataset with uniform polarity, the initial Euler angles from the subvolumes of opposite polarity were rotated by 180° around the *x*-axis. 409,896 subtomograms were extracted with a binning factor of 2 (6.84 Å per pixel) using a box size of $64^3$ pixels$^3$. The 20.7 Å de novo structure of F-actin in complex with Tpm (Supplementary Fig. 6b) was low-pass filtered and used as an initial reference. Global alignment of the in-plane angle was followed by several rounds of successive local refinement of the in-plane angle and all Euler angles. Simultaneously, shifts were refined within ±1.38 nm along the filament centerline. Once all the positions were refined, distance thresholding was used to remove oversampled positions, keeping only one subtomogram per actin subunit. 160,617 unbinned subvolumes were extracted using a box size of $128^3$ pixels$^3$. The dataset was split into two half sets that were refined independently.

To uncover potential changes in the azimuthal position of the Tpm strands between sarcomeres, each tomogram was then processed separately using the initial reference without Tpm density. All tomograms provided a comparable F-actin-Tpm structure, with the exception of the data shown in Fig. 4d in which the Tpm density was significantly shifted on F-actin. This indicates that most of the sarcomeres we captured were in a similar state of the contraction cycle. The three tomograms in which a M-line was detected were then further processed. To correlate the wide overlap observed at ML1 and ML2 with the structure of the thin filament, the 56,858 particles from these two tomograms were merged and further refined. The final maps associated with the different M-line organizations were sharpened, weighted by the FSC, and filtered to their respective resolution ($FSC_{0.143}$ = 15.7 and 17.7 Å for the myosin-state and intermediate-state structures, respectively; Supplementary Fig. 9c and Supplementary Table 2). All maps were visualized using UCSF Chimera or UCSF ChimeraX[71]. Map segmentation was performed using the Color Zone tool in Chimera or ChimeraX. Analysis of the distances and in-plane rotation between refined subvolumes belonging to the same filament permitted to determine the mean rise and angular twist of the F-actin structure.

Several pseudoatomic models containing F-actin in complex with Tpm in distinct functional states[10,19] were docked into the maps using rigid-body fitting in UCSF Chimera. The fit was limited to the actin region to reveal the functional state matching the Tpm density best.

**Reporting Summary**. Further information on research design is available in the Nature Research Reporting Summary linked to this article.

## Data availability
Data supporting the finding of this manuscript are available from the corresponding author upon reasonable request. One representative tomogram has been deposited in the EMDB under accession code EMD-12572. The in situ subtomogram averages of the sarcomeric actin-Tpm filament from neonatal Wistar rat cardiomyocytes have been deposited under accession codes EMD-11826 (for the myosin state) and EMD-11825 (for the intermediate state). Source data are provided with this paper.

## Code availability
The TOMOMAN package for the preprocessing of tomography data is available at: https://github.com/williamnwan/TOMOMAN. The deconvolution filter for tomograms is available at: https://github.com/dtegunov/tom_deconv. The NovaCTF 3D-CTF correction software for electron microscopy is available at: https://github.com/turonova/novaCTF. The tomosegmemtv software for membrane segmentation is available at: https://github.com/anmartinezs/pyseg_system/tree/master/code/tomosegmemtv. The STOPGAP package for subtomogram averaging is available at: https://github.com/williamnwan/STOPGAP.

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

## Acknowledgements

The authors thank W. Wan, F. Beck, P. S. Erdmann, S. Khavnekar, and V. Lucic for technical and computational assistance, and R. Poincloux for critical reading of the manuscript and helpful comments. This work was supported by the Max Planck Society and by the German Center for Cardiovascular Research (DZHK)-Munich Partner Site (to R. T. B.).

## Author contributions

L.B., J.S., and M.J. designed the research and interpreted the results. S.S. prepared the cells and performed immunofluorescence imaging. L.B. performed cryo-FIB milling, cryo-ET, tomogram reconstruction, and segmentation. L.B., J.S., and M.J. performed filament packing analysis. J.S. developed and applied the subtomogram averaging and polarity assessment approaches. R.T.B., W.B., P.S., and J.M.P. provided financial support and access to instrumentation. M.J. supervised the work and wrote the paper with contributions from the other authors.

## Funding

## Ethics declarations

Housing and use of laboratory animals at the Max Planck Institute of Biochemistry are fully compliant with all applicable German (e.g., German Animal Welfare Act) and EU (e.g., Annex III of Directive 2010/63/EU on the protection of animals used for scientific purposes) laws and regulations concerning care and use of laboratory animals. All of the animals were handled according to the approved license (No.5.1-568—rural districts office).

## Competing interests

The authors declare no competing interests.

**Additional information**

