## [Peer Review File · Nature Communications]

REVIEWER COMMENTS

Reviewer #1 (Remarks to the Author):

Review for “Molecular-scale visualization of sarcomere contraction within native cardiomyocytes” by Burbaum L, et al.

In this manuscript the authors have performed cryo-ET and subtomogram averaging on vitrified neonatal rat cardiomyocytes with the aim to describe the structural organization within the sarcomere.

Specifically, the authors have used segmentation and subtomogram averaging to determine the structure of the native cardiac thin filament in the sarcomere (exclusively the F-actin-Tpm complex) and to correlate thin filament polarity with respect to their position in the sarcomere. Overall, this is an interesting descriptive paper, with impressive in situ structural data of the sarcomere in a specific stage of cardiac development. To my understanding, this work is also the first to describe the sarcomere structure within cells using cryo-ET, in contrast to other recent work that has been performed on either the entire sarcomere of isolated mouse psoas muscle myofibrils (<https://doi.org/10.1101/2020.09.13.295386>) or more specifically on the Z-disk of isolated porcine cardiac myofibrils (<https://doi.org/10.1038/s42003-020-01321-5>).

The work of Burbaum L et al extends these insights into sarcomere organization. However, the manuscript would benefit from a changed focus on how the results are discussed.

The ultrastructural description of the sarcomere focuses almost exclusively on the A-band, not discussing in detail other structural features of the sarcomere. In addition, the structure of the sarcomere is described for a specific developmental stage in sarcomere formation, where there appears to be less order. Hence, this description requires a more detailed comparison how the presented findings relate to other work on sarcomere structures in later stages. Specifically, the authors should elaborate in more detail on their used model system and unique characteristics not present in other studies (such as the other previously published work). The authors have chosen neonatal cells as adult cells are too thick for vitrification via plunge freezing. This offers the potential to address and speculate in more detail about structural changes occurring upon development of cardiac muscle.

While this provides interesting points for discussion, the reduced order of thin filaments (i.e. the less regular hexagonal packing described in this manuscript) is posing additional challenges for image processing. The authors could acknowledge this in their manuscript, to provide a better understanding to the reader that their model system, while facilitating vitrification, makes structural analysis more difficult.

Additional comments:

1) I would suggest reformatting of figures to remove redundancy between some of the main and supplemental figures. Specifically, I suggest combining Figure 1 and Figure S1. The schematic representation in Figure S1a helps the reader, and otherwise the data given in S1 is redundant to 1.

- 2) Figures containing tomographic slices (Figure 1d and S3) need to provide information on the thickness of the shown tomographic slices.
- 3) Figure 1e: In this figure it appears that there is a gap between the actin filaments in the Z-disk. Is this real or just a visualization/segmentation error?
- 4) It is not clear from the figures, which data that is presented is derived from VPP tomograms and which comes from defocused tomograms.
- 5) Figure 4 can be reordered, as currently panel 4a is referenced later in the text than panel 4e.
- 6) Page 3, line13-15: "They feature regularly spaced Z-disks and A-bands, with a sarcomere length of 1.8 ± 0.2 μm (Figure 1b, S1d), similar to the value we found in adult mouse cells." This statement is somewhat ambiguous, as there is no presentation of adult mouse cell data (i.e. immunofluorescence images). As the authors make the comparison between neonatal and adult cells, the data should be included for both. See also the comment above, about the more detailed comparison between neonatal and adult sarcomeres.
- 7) Page 7, line 6: The statement "The central part of the structure exhibits a local resolution of 11.4 Å (Figure S9c), allowing docking of a pseudoatomic model of the F-actin-Tpm complex into the map (Figure 4b, Supplementary 8 Video S5)" is misleading. The global resolution of 13.2 Å is already sufficient for accurate fitting, hence the increased local resolution is not required to allow docking.
- 8) Page 10, line 5: "Six to eight EM grids were placed in 35 mm Petri dishes and 300 μl of cell suspension was added ". How can 300 μl be added to a 35 mm dish and then be cultured for 2 days, this volume sounds very low? How do the authors ensure that the small volume does not evaporate?
- 9) The polarity analysis is an interesting approach. I would be interested to know if the confidence of polarity analysis is related to filament orientation with respect to the tilt axis.
- 10) Page 12, line 16: more details on the used NovaCTF parameters are required, for example if data has been corrected via phaseflipping or multiplication.
- 11) The manuscript should contain a table summarizing all data acquisition and also image processing statistics.
- 12) Judging from the provided validation reports, no half maps and FSC masks are deposited. These should be added to the deposition.

Reviewer #2 (Remarks to the Author):

This is an excellent paper that uses the latest technology in cryo-electron tomography to study the in situ molecular structure and 3D organization of the contractile filaments of beating cultured cardiac myocytes. The results largely support past studies of isolated filaments using conventional EM methods, but here approach the gold standard of making such observations in living cells. The paper is clearly (if densely) written, beautifully illustrated, and the methodology comprehensively presented. The main and supplementary illustrations and all videos are excellent and crucial to the

argument. I have no major criticisms, but suggest that the following minor comments would help clarify or correct certain aspects. I will use the following notation to indicate page and line number: page X, line Y is indicated as X.Y.

1. 2.21. I suggest adding Huxley 1968 (ref 30) here.
2. 3.16. Cell thickness rather than size would convey the issue better?
3. 3.22. Please mark the SR in Fig 1c and S2d.
4. 3.23. ...bands that merge... I see that what is meant is the Z-lines, but this description was obscure to me at first. It is a bit misleading to call these bands, as they are very narrow (at first I thought you meant A-bands). I realize these structures are not strictly lines either, but lines comes closer to capturing what is meant. Perhaps if you point to what you mean by "merging" this would clarify the point as well, and reference specifically Fig 1c.
5. 4.7-8. It's not exactly clear what is being indicated by SR in Fig 1d. Also please point to ribosomes and state how these were identified. I think these are indicated by the white and black arrowheads in Fig 1d, but this should be stated in the legend.
6. 4.9-17. I found this para hard to follow, again because of the ambiguity of electron-dense "bands". The appearance of the dense lines in Fig 1c immediately suggests (to me at least) that these *are* Z-lines, so it may help to start by making that point. This would then orient the reader for the rest of the description, where you can establish the Z-line identity for certain as you do in the rest of the paragraph. As it currently reads, when you mention electron-dense bands at the start of this para (4.9-10), I am thinking of A-bands, and that the irregular fuzzy structures described are thick filaments. It took me careful reconsideration to realize you meant Z-lines.
7. 4.12. Fig 1d should be 1e.
8. 4.16. ... did not observe any myosin bare zone depleted of thin filaments. Since the bare zones in the tomograms are not apparent, their position is being deduced from where the Z-lines are. So I think it would help the reader to mark the approximate positions of the assumed bare zones in figures 1d, S3d-f.
9. 4.20. 45.1 nm is in pretty good agreement with X-ray diffraction. Might be worth referencing Matsubara JMB 82, 527-536, 1974 for this (or later Tom Irving papers).
10. 4.20-23. Interpretation of the plots in the top row of Fig 2a is not immediately obvious. When pointing out in the text that the thick filaments are in a hexagonal lattice and the thin filaments are at the trigonal points, why not support this by referring to the roughly 6-fold and 3-fold symmetry of the left and middle plots of Fig 2 (top).
11. The right plot shows a slightly more even distribution of density in the inner circle. Does this represent the 6 (or more) irregularly spaced actin filaments around each myosin filament?
12. In the left and right plots, what is the meaning of the higher radius concentric circle at twice the distance of the inner circle? This cannot represent nearest neighbors.
13. Overall I think it would help readers if you could give an intuitive feel for the meaning of these plots, so readers don't have to go into too great detail themselves understanding the nuances and being required to read and understand ref 25. Perhaps some guidance could be given in the figure legend?
14. A critical point in describing these interfilament distance measurements and thin filament numbers would be to indicate where they are being made (in the single filament overlap region of the sarcomere or the double overlap region, near the M-line?) as that would affect interpretation.
15. 4.25. the 2:1 ratio reported in ref 26 I think is for vertebrate *skeletal* muscle. The text should add "skeletal".
16. 5.1-2. The authors seem to imply that the different filament packing observed is due to the muscle being cardiac rather than skeletal. But they are looking at *neonatal* cardiac muscle, where the mature sarcomere structure may not yet have been achieved. Maybe the difference is in

the age of the muscle rather than cardiac vs. skeletal. Although I don't know the literature, I am sure there must be studies showing EM of the hexagonal lattice in cardiac muscle. It would be useful to compare with such studies. Our own lab's cardiac work shows a pretty regular thick filament lattice with 6 thin filaments around each thick filament at the trigonal points, similar to skeletal. A quick Google search shows similar images. I suggest that the additional thin filaments seen at the non-trigonal positions in this work are due to the immaturity of the muscle as much as (or more than) its being cardiac.

17. 5.4-5. ...suggests that... The logic here is not clear to me.

18. 5.23. In Fig 3b arrows have been used to represent the polarity of the thin filaments in terms of their barbed and pointed ends. It is a little confusing that the arrows used to represent this polarity have the opposite orientation to the pointed and barbed ends of the filaments. I.e. the back end (barbed end) of the arrow points toward the pointed end of the filament, and vice versa (see also 15.18, 24.3, 36.12, 36.19). If it is not a lot of trouble I think it would avoid sowing confusion in the reader's mind if the arrows were reversed, so that the barbed and pointed ends of the arrows corresponded respectively to the barbed and pointed ends of the filaments.

19. 6.2. I suggest referring also to the insets ML1-3 after "d, left".

20. 6.11. should "resting" be inserted before "value"?

21. 6.14.no clear overlap... To me there is clear overlap, but it is less.

22. 6.15. In referencing Video S4, I suggest indicating that it is the second half of the video, as the first half does not illustrate what is being discussed here.

23. 6.18. It might be clearer to sayto the different observed M-line organizations....

24. 6.21. ... a similar structure... Similar to what? Do you mean that thin filaments on opposite sides of the M-line are similar to each other?

25. 7.4. I think 5.9 nm is meant, not 35.9 nm.

26. 7.13-19. Nice interpretation. Has this difference been demonstrated in more than one sarcomere?

27. 8.4.demonstrates....

28. 10.10. Was this standard 2-sided blotting?

29. 11.19. Please spell out Volta phase plate (assuming this is what VPP is).

30. The figure labels often do not stand out against the background. It would help the reader to fix this.

31. Fig. 1. I found it a little difficult to follow the legend. Some parts of the figure are not specifically labeled (right image of b, right 2 images of d). Also, the use of the term inset to mean these zoomed-in areas is not exactly accurate, and leads to a bit more ambiguity, as insets would normally be on top of the original image, not shown to its right.

32. 23.4.of the idealized [??] array....?

33. Fig 4d. It would be easier to read the PDB codes if they were shown in stronger color.

34. Fig S1. Indicate that the samples are labeled with antibodies to the components shown in the top images.

35. 28.4. Mitochondrion.

36. Fig S4. Interpretation of this figure was obscure to me. I recognize the analysis is complex, as described in the methods, but it would help the reader if some kind of intuitive feel could be given as to how to interpret. I take it that the filament distances we are to focus on are the narrow part of the distribution at the left and lowest part of each histogram, representing regions of filament closest to parallel to their neighbors. What is represented by the much larger number of measurements in the large area of each panel to the right of this narrow range? These numbers go as high as 200 nm – do these truly represent nearest neighbors?

37. 31.9. Is difference map the correct term? The map at right does not appear to show the

difference between two structures but rather the superposition of them.

Reviewer #3 (Remarks to the Author):

The manuscript 'Molecular-scale visualization of sarcomere contraction withing native cardiomyocytes' by Burbaum, Schneider and colleagues describes an extensive analysis of the sarcomere organization in rat cardiomyocytes. The study is based on cryo-EM tomography datasets and upon reconstruction and segmentation of thin and thick filaments, myofilament packing is mapped. The authors determine the structural signature of sarcomere contraction.

1. While the manuscript is very well written and the analysis is performed thoroughly, I believe the manuscript would benefit from more extensive documentation of the experimental setup. It is not clear if different tomograms are derived from the same cell sample or not. Please add more info in the methods section, e.g.:

Methods line 6: Cryo FIB milling - How many cells/grids were used for making lamellae? Was it possible to specifically select contracting cells, based on their appearance in the SEM, or were cells chosen randomly?

Methods line 22: Cryo-ET - How many tilt series were collected? From how many different cells?

2. Please, add the controls of the immunofluorescence experiment to Suppl. Figure 1.

3. The organisation of the figures is confusion (especially figure 3). It might be more clear to have a unique letter for each individual panel. Also a more detailed description in the figure legends would make things more clear.

4. Methods line 19: VPP - abbreviation not explained.

5. Figure S10 shows your proposed model - this would be ideal as a final figure, not as supplementary material.

We thank the reviewers for their encouraging and valuable comments, and the editor for giving us the opportunity to revise the manuscript. We have addressed the reviewers' comments in the detailed point-by-point response provided below.

In short, we have emphasized that our results were obtained at a specific stage of cardiac development and have provided detailed comparisons with the adult stage. We have also speculated on the structural changes that occur upon development of cardiac muscle, especially regarding myofibril assembly and myofilament packing. The immunofluorescence data of the adult mouse cells have been included. In addition, we have provided more detail on the packing analysis in the figure legends, as well as on the experimental set up in the Methods and with two additional tables.

We have prepared a revised version of the manuscript with changes indicated in blue, which follows the guidelines of Nature Communications. We will use the same notation as Reviewer 2 to indicate page and line number: page X, line Y is indicated as X.Y.

Reviewers' comments

Reviewer #1 (Remarks to the Author):

Review for "Molecular-scale visualization of sarcomere contraction within native cardiomyocytes" by Burbaum L, et al.

In this manuscript the authors have performed cryo-ET and subtomogram averaging on vitrified neonatal rat cardiomyocytes with the aim to describe the structural organization within the sarcomere.

Specifically, the authors have used segmentation and subtomogram averaging to determine the structure of the native cardiac thin filament in the sarcomere (exclusively the F-actin-Tpm complex) and to correlate thin filament polarity with respect to their position in the sarcomere. Overall, this is an interesting descriptive paper, with impressive in situ structural data of the sarcomere in a specific stage of cardiac development. To my understanding, this work is also the first to describe the sarcomere structure within cells using cryo-ET, in contrast to other recent work that has been performed on either the entire sarcomere of isolated mouse psoas muscle myofibrils (<https://doi.org/10.1101/2020.09.13.295386>) or more specifically on the Z-disk of isolated porcine cardiac myofibrils (<https://doi.org/10.1038/s42003-020-01321-5>).

The work of Burbaum L et al extends these insights into sarcomere organization. However, the manuscript would benefit from a changed focus on how the results are discussed.

The ultrastructural description of the sarcomere focuses almost exclusively on the A-band, not discussing in detail other structural features of the sarcomere. In addition, the structure of the sarcomere is described for a specific developmental stage in sarcomere formation, where there appears to be less order. Hence, this description requires a more detailed comparison how the presented findings relate to other work on sarcomere structures in later stages. Specifically, the authors should elaborate in more detail on their used model system and unique characteristics not present in other studies (such as the other previously

published work). The authors have chosen neonatal cells as adult cells are too thick for vitrification via plunge freezing. This offers the potential to address and speculate in more detail about structural changes occurring upon development of cardiac muscle. While this provides interesting points for discussion, the reduced order of thin filaments (i.e. the less regular hexagonal packing described in this manuscript) is posing additional challenges for image processing. The authors could acknowledge this in their manuscript, to provide a better understanding to the reader that their model system, while facilitating vitrification, makes structural analysis more difficult.

We thank the reviewer for these encouraging comments and valuable suggestions. As requested, we have elaborated on our model system and have discussed our findings in relation to previous work on adult vertebrate cardiac muscles. As vertebrate cardiac and skeletal muscles share the same myofibril ultrastructure, we have also discussed vertebrate skeletal muscle studies whenever appropriate. A detailed description is provided below:

- Our model system has been introduced at the beginning of the results section together with the adult cell system (3.10-14). The immunofluorescence data of both cell types have been presented and discussed (3.18-25, 26.2-11, 32.1-9).

- The diameter and the microscale organization of the neonatal cardiac myofibrils have been compared with those of adult mammalian cardiac myofibrils imaged by EM (4.8-10 and 4.12-16).

- We have provided more information about the Z-disk: its zig-zag shaped morphology has been discussed and we have speculated on myofibril assembly upon development of cardiac muscle (5.3-12).

- We have compared the widths of our Z-disks and I-bands with those of adult vertebrate cardiac sarcomeres (5.17-20), and acknowledged that structural analysis is limited in these regions (5.16-17 and 5.20-22). We have referred to the recent cryo-ET studies on isolated myofibrils from adult vertebrate striated muscles (5.22-6.2).

- We have emphasized that we focused on the A-band organization (6.4-6). The packing in the A-bands has been compared with earlier work on adult vertebrate cardiac and skeletal muscles using X-ray diffraction and EM (6.13-15, 6.20-23). We have also speculated on the changes in the myofilament organization that occur upon vertebrate muscle development (6.23-7.6).

- The established polarity observed at the neonatal stage has been compared with studies on embryonic and adult mammalian skeletal muscles (8.1-3).

- We have elaborated on our M-lines results (8.7-9.7). The double overlap of the thin filaments observed in the neonatal cardiac M-lines has been discussed with previous findings on adult vertebrate skeletal and cardiac muscle fibers (8.11-20, 9.4-7). We have showed that the extent of the overlap agrees well with the measured sarcomere lengths (8.19-20, 9.4-7), and discussed the number of thin filaments observed at ML1-2 in relation to the variation in thin filament length in cardiac muscle (8.23-9.1).

- In our summary statement, we have emphasized that our work provided structural insights into a specific stage of cardiac development, while supporting past studies of adult vertebrate striated muscles (10.17-20). The reduced order of the thin filaments has been acknowledged (10.22-23), as well as the importance of further *in situ* studies on other cell systems at different developmental stages (11.4-6).

Additional comments:

1) I would suggest reformatting of figures to remove redundancy between some of the main and supplemental figures. Specifically, I suggest combining Figure 1 and Figure S1. The schematic representation in Figure S1a helps the reader, and otherwise the data given in S1 is redundant to 1.

We have moved the scheme from Fig. S1a to Fig. 1a (26.1-6). Since we also wanted to include the zoomed-in views of the immunofluorescence data, we have split Fig. 1 into two: new Fig. 1 contains the scheme and the immunofluorescence data of the neonatal cells while new Fig. 2 shows the TEM and cryo-ET data.

2) Figures containing tomographic slices (Figure 1d and S3) need to provide information on the thickness of the shown tomographic slices.

The thickness of the slices has been added to the legends of Fig. 2b (27.6) and Fig. S3a-c (34.3).

3) Figure 1e: In this figure it appears that there is a gap between the actin filaments in the Z-disk. Is this real or just a visualization/segmentation error?

The gap results from the high density in the Z-disk structure, which limits the segmentation of the thin filament ends. This information has been added to the main text (5.16-17).

4) It is not clear from the figures, which data that is presented is derived from VPP tomograms and which comes from defocused tomograms.

The figures have been prepared using the defocused tomograms. We have added this information to the figure legends (27.6, 34.3).

5) Figure 4 can be reordered, as currently panel 4a is referenced later in the text than panel 4e.

Fig. 4e has been moved to Fig. S9a. This panel has been extended to show the three structures obtained from the data presented in Fig. 4b-d (ML1-3) as well as their superpositions (ML1/2, ML1/3, ML2/3; 40.3-8).

In addition, to allow a direct correlation between the M-line organization and the thin filament structure, we selected the two tomograms with a visible M-line and a substantial overlap of thin filaments (Fig. 4b-c) to generate the myosin-state structure (Fig. 5a). The resolution of 15.7 Å (Fig S9c) allowed docking of pseudoatomic models in different states into the map (Fig. 5d and Fig S9d). The two structures in distinct functional states are now shown with their respective fits in Fig. 5 (30.1-13). Fig. S9 and Movie S5 have been updated. Furthermore, as suggested by Reviewer 3, our proposed model (old Fig. S10) has been moved to Fig. 6.

6) Page 3, line13-15: “They feature regularly spaced Z-disks and A-bands, with a sarcomere length of 1.8 ± 0.2 μm (Figure 1b, S1d), similar to the value we found in adult mouse cells.” This statement is somewhat ambiguous, as there is no presentation of adult mouse cell data (i.e. immunofluorescence images). As the authors make the comparison between neonatal and adult cells, the data should be included for both. See also the comment above, about the more detailed comparison between neonatal and adult sarcomeres.

We agree with the reviewer that this statement was misleading since the immunofluorescence data of the adult cells was not shown. The data of the adult mouse cells has been added to Fig. S1 and have been discussed in the results section (3.23-25). In addition, our statement has been corrected. Indeed, the sarcomere length measured in the adult mouse cells is 1.5 ± 0.1 μm , which is close but different from that found in the neonatal rat cardiomyocytes. As these small variations in sarcomere length could be due to differences between species, we have left out the comparison to avoid any misinterpretation (32.7-9).

7) Page 7, line 6: The statement “The central part of the structure exhibits a local resolution of 11.4 Å (Figure S9c), allowing docking of a pseudoatomic model of the F-actin-Tpm complex into the map (Figure 4b, Supplementary 8 Video S5)” is misleading. The global resolution of 13.2 Å is already sufficient for accurate fitting, hence the increased local resolution is not required to allow docking.

We thank the reviewer for this comment. As explained in the response to comment 5), the structure generated from the two tomograms with a visible M-line and a substantial overlap of thin filaments has a resolution of 15.7 Å. This is sufficient for accurate fitting and we have amended the sentence as follows (10.2-3):

“The resolution allows docking of a pseudoatomic model of the F-actin-Tpm complex into the map”.

8) Page 10, line 5: “Six to eight EM grids were placed in 35 mm Petri dishes and 300 μl of cell suspension was added”. How can 300 μl be added to a 35 mm dish and then be cultured for 2 days, this volume sounds very low? How do the authors ensure that the small volume does not evaporate?

The sentence has been corrected in the Methods (13.5-7):

“Six to eight EM grids were placed in 35 mm Petri dishes containing 1.5 mL of cardiomyocyte medium and 300 μL of cell suspension was added”

9) The polarity analysis is an interesting approach. I would be interested to know if the confidence of polarity analysis is related to filament orientation with respect to the tilt axis. 74% of the filaments in our data have an orientation relative to the tilt axis within the $[15^\circ; 45^\circ]$ range, with values up to 70° . We have compared the angular distributions between the assigned and unassigned filaments and did not observe any significant differences between them.

In contrast, there is a strong dependence with filament length. Assigned filaments are on average twice as long as unassigned filaments (with mean lengths of 470 and 250 nm, respectively).

10) Page 12, line 16: more details on the used NovaCTF parameters are required, for example if data has been corrected via phaseflipping or multiplication.

The following information has been included in the Methods (15.22-24):

“3D-CTF correction was performed using NovaCTF⁶⁴ with phaseflip correction and a defocus step of 15 nm”

11) The manuscript should contain a table summarizing all data acquisition and also image processing statistics.

We have added two tables (42.1-5): Table S1 summarizes the data acquisition parameters while Table S2 summarizes image processing statistics. References to these tables have been added to the main text (4.17-18, 9.23, 10.12) and Methods (15.8-9, 16.15, 17.14, 20.5-6).

12) Judging from the provided validation reports, no half maps and FSC masks are deposited. These should be added to the deposition.

The EMBD entry for the myosin state structure has been updated. As recommended, the half maps and FSC masks for both structures have been deposited (see new validation reports).

Reviewer #2 (Remarks to the Author):

This is an excellent paper that uses the latest technology in cryo-electron tomography to study the in situ molecular structure and 3D organization of the contractile filaments of beating cultured cardiac myocytes. The results largely support past studies of isolated filaments using conventional EM methods, but here approach the gold standard of making such observations in living cells. The paper is clearly (if densely) written, beautifully illustrated, and the methodology comprehensively presented. The main and supplementary illustrations and all videos are excellent and crucial to the argument. I have no major criticisms, but suggest that the following minor comments would help clarify or correct certain aspects. I will use the following notation to indicate page and line number: page X, line Y is indicated as X.Y.

We thank the reviewer for these encouraging and valuable comments. As suggested, the relevant parts of the manuscript have been clarified or corrected.

1. 2.21. I suggest adding Huxley 1968 (ref 30) here.

The reference has been added.

2. 3.16. Cell thickness rather than size would convey the issue better?

We agree and have modified it in the text.

3. 3.22. Please mark the SR in Fig 1c and S2d.

The SR identified in the tomogram shown in Fig. 2b (old Fig. 1d) has been marked in Fig. 2a (old Fig. 2c) and Fig. S2d.

4. 3.23. ...bands that merge... I see that what is meant is the Z-lines, but this description was obscure to me at first. It is a bit misleading to call these bands, as they are very narrow (at first I thought you meant A-bands). I realize these structures are not strictly lines either, but

lines comes closer to capturing what is meant. Perhaps if you point to what you mean by “merging” this would clarify the point as well, and reference specifically Fig 1c.

We thank the reviewer for this suggestion. We have replaced “bands” by “lines” in the main text (4.10, 4.12, 5.1) and in the legend of Fig. 2 (27.4).

5. 4.7-8. It’s not exactly clear what is being indicated by SR in Fig 1d. Also please point to ribosomes and state how these were identified. I think these are indicated by the white and black arrowheads in Fig 1d, but this should be stated in the legend.

In Fig. 2b (old Fig. 1d), “SR” has been moved up to allow better visibility of the SR membranes. It is now indicated in the legend of Fig. 2 (27.7-8) that ribosomes and glycogen granules are indicated by white and black arrowheads, respectively (see also between panels **b** and **c** of Fig. 2). Ribosomes are commonly observed in tomograms. They were identified by their shape and size (~25 nm in diameter). The white arrowheads have been adjusted to precisely point to one ribosome each.

6. 4.9-17. I found this para hard to follow, again because of the ambiguity of electron-dense “bands”. The appearance of the dense lines in Fig 1c immediately suggests (to me at least) that these *are* Z-lines, so it may help to start by making that point. This would then orient the reader for the rest of the description, where you can establish the Z-line identity for certain as you do in the rest of the paragraph. As it currently reads, when you mention electron-dense bands at the start of this para (4.9-10), I am thinking of A-bands, and that the irregular fuzzy structures described are thick filaments. It took me careful reconsideration to realize you meant Z-lines.

The word “bands” has been replaced by “lines”. As suggested, we also immediately stated that these electron-dense lines are Z-disks (5.1-3):

“The periodic, electron-dense lines observed in the TEM images are made of irregular, densely packed structures that intersect with the thin filaments in the absence of thick filaments, confirming that they are Z-disks”

7. 4.12. Fig 1d should be 1e.

This has been corrected and now corresponds to Fig. 2e (old Fig. 1e).

8. 4.16. ... did not observe any myosin bare zone depleted of thin filaments. Since the bare zones in the tomograms are not apparent, their position is being deduced from where the Z-lines are. So I think it would help the reader to mark the approximate positions of the assumed bare zones in figures 1d, S3d-f.

The putative myosin bare zones have been added to Fig. 2f, g and Fig. S3h, i, k, l based on the Z-disk position and the average sarcomere length of 1.8 μm . They are mentioned in the text in 6.6-8.

9. 4.20. 45.1 nm is in pretty good agreement with X-ray diffraction. Might be worth referencing Matsubara JMB 82, 527-536, 1974 for this (or later Tom Irving papers).

We thank the reviewer for pointing out these references. They have been added to the text as follows (6.13-15):

“Given the sarcomere lengths measured in our cells, these values are in good agreement with the spacings found in adult mammalian heart muscle using X-ray diffraction^{40,41}.”

10. 4.20-23. Interpretation of the plots in the top row of Fig 2a is not immediately obvious. When pointing out in the text that the thick filaments are in a hexagonal lattice and the thin filaments are at the trigonal points, why not support this by referring to the roughly 6-fold and 3-fold symmetry of the left and middle plots of Fig 2 (top).

As suggested, the revised text has been clarified as follows (6.11-18):

“As shown by the six-fold symmetry in Fig. 3a, thick filaments organize in a hexagonal lattice with an interfilament distance of 45.1 ± 3.8 nm (Fig. 3a-c and Supplementary Fig. 4a, b). [...] The roughly three-fold symmetry in Fig. 3d indicates that the thin filaments are found at the trigonal positions of the lattice, with a distance of 26.0 ± 2.4 nm from the thick filaments (Fig. 3d-f and Supplementary Fig. 4c-f). Combined together, these spacings agree precisely with the geometrical constraints of a double hexagonal lattice.”

11. The right plot shows a slightly more even distribution of density in the inner circle. Does this represent the 6 (or more) irregularly spaced actin filaments around each myosin filament?

Yes. We have added this information to the legend of Fig. 3 (old Fig. 2; 28.16-19):

“They are also located outside the trigonal positions of the thick-filament lattice (g-i), as indicated by the more even distribution of density in the first shell in g and the distance of about 15.5 nm between nearest parallel thin filaments (Supplementary Fig. 4g, h).”

12. In the left and right plots, what is the meaning of the higher radius concentric circle at twice the distance of the inner circle? This cannot represent nearest neighbors.

They correspond to parallel neighboring filaments in the second shell around the filaments, that is, at twice the distance determined for the nearest neighbors. The nearest neighbors are within the smaller concentric circle.

13. Overall I think it would help readers if you could give an intuitive feel for the meaning of these plots, so readers don't have to go into too great detail themselves understanding the nuances and being required to read and understand ref 25. Perhaps some guidance could be given in the figure legend?

We agree that this analysis is complex and thank the reviewer for this suggestion. We have added additional information in the legends of Fig. 3 (28.7-14) and Fig. S4 (35.7-16).

14. A critical point in describing these interfilament distance measurements and thin filament numbers would be to indicate where they are being made (in the single filament overlap region of the sarcomere or the double overlap region, near the M-line?) as that would affect interpretation.

We thank the reviewer for this valuable point. The quantitative analysis has been performed on the entire A-band region. Given that the polarity analysis only revealed a few double-overlap regions over less than 20% of the A-band length, their contribution to the packing analysis can be considered to be small. Therefore, the interfilament distances and the myofilament organization are representative of the single-overlap region. We have added this information to the Results (6.9-11) and Methods (16.11-14). In addition, Fig. S5b has been updated so that all the real cross-sections in Figs. 3 and S5 are well within the single-overlap region. This has also been stated in the legends (28.5 and 36.3). To avoid confusion, we have also removed the cross-section through the M-line from our proposed model (old Fig. S10, new Fig. 6 as suggested by Reviewer 3).

The limited number of double-overlap regions we found does not allow for quantitative analysis in this region. Interestingly, the cross-sections through ML1-2 show no significant increase in the number of thin filaments around the thick filaments in this region (Fig. S8g,h). This could be explained by the variation in thin filament length within a single sarcomere found in cardiac muscle, which results in incomplete shells of thin filaments around thick filament near the M-region⁴⁹. This discussion point has been added to the polarity results section (8.23-9.1).

15. 4.25. the 2:1 ratio reported in ref 26 I think is for vertebrate *skeletal* muscle. The text should add "skeletal".

This has been added (6.25). In addition, we have also included a reference for vertebrate cardiac muscle (see next response for more information).

16. 5.1-2. The authors seem to imply that the different filament packing observed is due to the muscle being cardiac rather than skeletal. But they are looking at *neonatal* cardiac muscle, where the mature sarcomere structure may not yet have been achieved. Maybe the difference is in the age of the muscle rather than cardiac vs. skeletal. Although I don't know the literature, I am sure there must be studies showing EM of the hexagonal lattice in cardiac muscle. It would be useful to compare with such studies. Our own lab's cardiac work shows a pretty regular thick filament lattice with 6 thin filaments around each thick filament at the trigonal points, similar to skeletal. A quick Google search shows similar images. I suggest that the additional thin filaments seen at the non-trigonal positions in this work are due to the immaturity of the muscle as much as (or more than) its being cardiac.

We thank the reviewer for this comment and fully agree that the reduced order of the thin filament is related to the neonatal stage. As we were only referring to previous studies on skeletal muscle, our statement was misleading. We have included the following reference: "Stenger, R. J. & Spiro, D. The ultrastructure of mammalian cardiac muscle. *J Biophys Biochem Cytol* (1961)" to clarify that vertebrate cardiac and skeletal muscles share the same ultrastructure, with the same 2:1 ratio at the adult stage, and emphasize the neonatal versus adult comparison (6.20-7.1). In addition, we have broadened the discussion by speculating on the structural changes occurring during vertebrate muscle development (7.1-4).

17. 5.4-5. ...suggests that... The logic here is not clear to me.

We have removed this speculative statement. The text now states (7.4-6):

"In addition, our data show that sarcomere contraction can occur despite the higher density and reduced order of the thin filaments around the hexagonally packed thick filaments."

18. 5.23. In Fig 3b arrows have been used to represent the polarity of the thin filaments in terms of their barbed and pointed ends. It is a little confusing that the arrows used to represent this polarity have the opposite orientation to the pointed and barbed ends of the filaments. I.e. the back end (barbed end) of the arrow points toward the pointed end of the filament, and vice versa (see also 15.18, 24.3, 36.12, 36.19). If it is not a lot of trouble I think it would avoid sowing confusion in the reader's mind if the arrows were reversed, so that the barbed and pointed ends of the arrows corresponded respectively to the barbed and pointed ends of the filaments.

We thank the reviewer for pointing that out. We have changed the direction of the arrows in Fig. 4 (old Fig. 3), Fig. S8 and Movies S3-S4, and modified the legends accordingly (29.3, 39.7, 41.12, 41.20).

19. 6.2. I suggest referring also to the insets ML1-3 after “d, left”.
This has been updated as “Fig. 4f-h, ML1-3” (8.5).

20. 6.11. should “resting” be inserted before “value”?

The neonatal cardiomyocytes were beating prior to the immunofluorescence labelling experiment. Therefore, the 1.8- μm value represents an average between contracting and relaxing cells. This is also in agreement with the values measured for the sarcomeres in the myosin (1.65 μm) and intermediate (1.96 μm) state. We have added “average” before “value” (8.18).

21. 6.14.no clear overlap... To me there is clear overlap, but it is less.

We thank the reviewer for this comment. We have measured an overlap length of 60 nm, corresponding to 3% of the sarcomere length. We have adjusted the text as follows (9.1-7): “In the sarcomere shown in Fig. 4d, the overlap between thin filaments of opposite polarity is barely visible, indicating that this sarcomere may be in a different contraction state (Fig. 4h, i, ML3, Supplementary Fig. 8f, i, and second half of Supplementary Movie 4). We measured an overlap length of 60 nm for a sarcomere length of 1.96 μm , i.e. an overlap region of only 3% of the sarcomere length. This agrees with the onset of thin filament overlap estimated for a sarcomere length of about 2 μm in adult frog skeletal muscle fibers⁴⁸.”

22. 6.15. In referencing Video S4, I suggest indicating that it is the second half of the video, as the first half does not illustrate what is being discussed here.

We now refer to the first (8.11) and second (9.3-4) halves of Movie S4 specifically.

23. 6.18. It might be clearer to sayto the *different* observed M-line organizations....

We have updated the text (9.11).

24. 6.21. ... a similar structure... Similar to what? Do you mean that thin filaments on opposite sides of the M-line are similar to each other?

We meant that the F-actin-Tpm structures obtained for these tomograms were similar.

This part of the results has been changed: we selected the three tomograms in which an M-line has been observed to obtain the thin filament structures (see also answer to comment 5) of Reviewer 1). This enabled us to confidently correlate M-line organization with functional state.

We have rephrased the sentence as (9.13-15):

“The two tomograms where thin filaments of opposite polarity substantially overlap in the M-line region provided a similar F-actin-Tpm structure (Supplementary Fig. 9a, blue (ML1) and dark blue (ML2) Tpm densities, ML1/2).”

In addition, we now state (9.19-23):

“To correlate the wide overlap observed at the M-lines in Fig. 4b, c (ML1-2) with the structure of the thin filament, the particles from these two tomograms were combined. A refined subtomogram average of the cardiac thin filament at a resolution of 15.7 Å (Fig. 5a,

Supplementary Fig. 9b-c (orange Tpm/curve), Supplementary Movie 5 and Supplementary Table 2).”

25. 7.4. I think 5.9 nm is meant, not 35.9 nm.

We meant the full repeat distance of the single helix. “Pitch” has been replaced by “full repeat distance” (10.1).

26. 7.13-19. Nice interpretation. Has this difference been demonstrated in more than one sarcomere?

The data shown in Fig. 4d (old Fig. 3d) contain two adjoining sarcomeres, both of which were used to generate the intermediate-state structure. This is why we had stated: “these sarcomeres are in a different state”. Since only one of them has a visible M-line (ML3), we have modified the text to: “this sarcomere is in a different state” (10.13).

27. 8.4.demonstrates....

This has been corrected.

28. 10.10. Was this standard 2-sided blotting?

Cells were blotted from the back side only. This information has been added to the Methods (13.12).

29. 11.19. Please spell out Volta phase plate (assuming this is what VPP is).

This has been spelled out (14.21).

30. The figure labels often do not stand out against the background. It would help the reader to fix this.

This has been fixed.

31. Fig. 1. I found it a little difficult to follow the legend. Some parts of the figure are not specifically labeled (right image of **b**, right 2 images of **d**). Also, the use of the term inset to mean these zoomed-in areas is not exactly accurate, and leads to a bit more ambiguity, as insets would normally be on top of the original image, not shown to its right.

The zoomed-in views are now labelled as Fig. 2c,d and correctly named (27.10, 27.12).

32. 23.4.of the *idealized* [??] array....?

This has been added (28.4).

33. Fig 4d. It would be easier to read the PDB codes if they were shown in stronger color.

They have been put in bold in Figs. 5 and S9.

34. Fig S1. Indicate that the samples are *labeled with antibodies* to the components shown in the top images.

The legends of Figs. 1 and S1 have been updated accordingly (26.6-7, 34.3-4).

35. 28.4. Mitochondrion.

This has been corrected (27.5, 33.10, 34.5).

36. Fig S4. Interpretation of this figure was obscure to me. I recognize the analysis is complex, as described in the methods, but it would help the reader if some kind of intuitive feel could be given as to how to interpret. I take it that the filament distances we are to focus on are the narrow part of the distribution at the left and lowest part of each histogram, representing regions of filament closest to parallel to their neighbors. What is represented by the much larger number of measurements in the large area of each panel to the right of this narrow range? These numbers go as high as 200 nm – do these truly represent nearest neighbors?

The analysis does not look for nearest-neighbors exclusively. It searches for the closest point of each neighboring filament within a certain distance range. The peak occurring at the shortest interfilament distances and small θ (below 10°) corresponds to the contribution of the nearest parallel filaments. The other measurements at larger distances correspond to contributions from neighboring filaments within the distance range.

To clarify this point, we have changed the analysis name to “near-neighbor analysis” (16.9, 16.19, 35.3) and provided more information in the figure legend (35.7-16).

37. 31.9. Is difference map the correct term? The map at right does not appear to show the difference between two structures but rather the superposition of them.

We have corrected the sentence as follows (37.9-12):

“d Filament structures in opposite orientations (green and purple, respectively) and aligned on the Tpm densities. e Respective difference maps with superimposed Tpm densities (grey) revealing the actin positions associated to each polarity with respect to the Tpm.”

Reviewer #3 (Remarks to the Author):

The manuscript 'Molecular-scale visualization of sarcomere contraction withing native cardiomyocytes' by Burbaum, Schneider and colleagues describes an extensive analysis of the sarcomere organization in rat cardiomyocytes. The study is based on cryo-EM tomography datasets and upon reconstruction and segmentation of thin and thick filaments, myofilament packing is mapped. The authors determine the structural signature of sarcomere contraction.

1. While the manuscript is very well written and the analysis is performed thoroughly, I believe the manuscript would benefit from more extensive documentation of the experimental setup. It is not clear if different tomograms are derived from the same cell sample or not. Please add more info in the methods section, e.g.:

Methods line 6: Cryo FIB milling - How many cells/grids were used for making lamellae? Was it possible to specifically select contracting cells, based on their appearance in the SEM, or were cells chosen randomly?

Methods line 22: Cryo-ET - How many tilt series were collected? From how many different cells?

We thank the reviewer for these encouraging comments and valuable suggestions. We have added the following information to the Methods:

- “A total of eight lamellas from randomly chosen cells was used to produce the data presented in the paper” (cryo-FIB section, 14.22-23)

- “A total of thirteen tomograms was used in this study (Supplementary Table 1). Three of the tilt series, collected from one cell, were recorded unidirectionally with the VPPs at a target defocus of 0.5 μm . Ten of the tilt series, collected from seven different cells, were recorded without the VPP using a dose-symmetric tilt-scheme⁵⁷ and a target defocus range of -3.25 to -5 μm .” (cryo-ET section, 15.8-12).

We have also summarized our data acquisition parameters and image processing statistics in two additional tables (Tables S1-S2, 42.1-5).

2. Please, add the controls of the immunofluorescence experiment to Suppl. Figure 1. The immunofluorescence data for the adult mouse cells have been presented in Fig. S1 (32.1-9) and discussed in the main text (3.23-25).

3. The organisation of the figures is confusion (especially figure 3). It might be more clear to have a unique letter for each individual panel. Also a more detailed description in the figure legends would make things more clear.

We thank the reviewer for pointing that out. A unique letter is now used for each individual panel. We have also extended the figure legends, especially for the packing and polarity results.

4. Methods line 19: VPP - abbreviation not explained.
“Volta phase plate” has been added (14.21).

5. Figure S10 shows your proposed model - this would be ideal as a final figure, not as supplementary material.

We thank the reviewer for this suggestion. We have moved Fig. S10 to the main text as Fig. 6 (31.1-6). To support our model further, we selected the tomograms with a visible M-line to obtain the thin filament structures.

Additional comments

- The Tpn subunit labelled with antibodies is Tpn T (not Tpn I). This has been corrected in the text (3.16) and figure legends (26.7, 34.3).

- The percentage provided for the polarity assessment corresponds to the percentage of assigned particles (not filaments). This has been corrected in the results section (7.22), Methods (19.4) and legend of Fig. S7 (38.12) and Fig. S8 (39.5).

- The original tomogram shown in Fig. 2 and Movie S2 has been deposited in the EMDB under accession code EMD-12572.

- We added to the Methods that the TOMOMAN package was used for the preprocessing steps of the tomogram reconstruction (15.15-16).

REVIEWERS' COMMENTS

Reviewer #1 (Remarks to the Author):

The revisions have significantly improved the manuscript. I am happy to recommend it for publication.

Reviewer #2 (Remarks to the Author):

This is a very nice paper. The authors have carried out a thorough revision, responding appropriately to all of my suggestions. The presentation, already good, is now significantly clearer. I believe they have also responded well to the other reviewers' suggestions. In my opinion, this paper is now ready for publication.

Reviewer #3 (Remarks to the Author):

Dear authors,

thank you for addressing my questions. Your edits have resulted in a very clear and valuable manuscript.

Best regards,

Saskia Lippens